# Exponentially convergent stochastic $k$-PCA without variance reduction

Cheng Tang

Amazon AI *
New York, NY, 10001
tcheng@amazon.com

## Abstract

We present *Matrix Krasulina*, an algorithm for online $k$-PCA, by generalizing the classic Krasulina's method [1] from vector to matrix case. We show, both theoretically and empirically, that the algorithm naturally adapts to data low-rankness and converges exponentially fast to the ground-truth principal subspace. Notably, our result suggests that despite various recent efforts to accelerate the convergence of stochastic-gradient based methods by adding a $O(n)$-time variance reduction step, for the $k$-PCA problem, a truly online SGD variant suffices to achieve exponential convergence on intrinsically low-rank data.

## 1 Introduction

Principal Component Analysis (PCA) is ubiquitous in statistics, machine learning, and engineering alike: For a centered $d$-dimensional random vector $X \in \mathbb{R}^d$, the $k$-PCA problem is defined as finding the "optimal" projection of the random vector into a subspace of dimension $k$ so as to capture as much of its variance as possible; formally, we want to find a rank $k$ matrix $W$ such that

$$\max_{W \in \mathbb{R}^{k \times d}, WW^\top = I_k} \text{Var} \left( W^\top W X \right)$$

In the objective above, $W^\top W = W^\top (WW^\top)^{-1} W$ is an orthogonal projection matrix into the subspace spanned by the rows of $W$. Thus, the $k$-PCA problem seeks matrix $W$ whose row-space captures as much variance of $X$ as possible. This is equivalent to finding a projection into a subspace that minimizes variance of data outside of it:

$$\min_{W \in \mathbb{R}^{k \times d}, WW^\top = I_k} \mathbb{E} \, \|X - W^\top W X\|^2 \tag{1.1}$$

Likewise, given a sample of $n$ centered data points $\{X_i\}_{i=1}^n$, the empirical version of problem (1.1) is

$$\min_{W \in \mathbb{R}^{k \times d}, WW^\top = I_k} \frac{1}{n} \sum_{i=1}^n \|X_i - W^\top W X_i\|^2 \tag{1.2}$$

The optimal $k$-PCA solution, the row space of optimal $W$, can be used to represent high-dimensional data in a low-dimensional subspace ($k \ll d$), since it preserves most variation from the original data. As such, it usually serves as the first step in exploratory data analysis or as a way to compress data before further operation.

The solutions to the nonconvex problems (1.1) and (1.2) are the subspaces spanned by the top $k$ eigenvectors (also known as the *principal subspace*) of the population and empirical data covariance

matrix, respectively. Although we do not have access to the population covariance matrix to directly solve (1.1), given a batch of samples $\{x_i\}_{i=1}^n$ from the same distribution, we can find the solution to (1.2), which asymptotically converges to the population $k$-PCA solution [2]. Different approaches exist to solve (1.2) depending on the nature of the data and the computational resources available:

**SVD-based solvers** When data size is manageable, one can find the exact solution to (1.2) via a singular value decomposition (SVD) of the empirical data matrix in $\min\{O(nd^2), O(n^2 d)\}$-time and $O(nd)$-space, or in case of truncated SVD in $O(ndk)$-time ($O(nd \log k)$ for randomized solver [3]).

**Power method** For large-scale datasets, that is, both $n$ and $d$ are large, the full data may not fit in memory. Power method [4, p.450] and its variants are popular alternatives in this scenario; they have less computational and memory burden than SVD-based solvers; power method approximates the principal subspace iteratively: At every iteration, power method computes the inner product between the algorithm's current solution and $n$ data vectors $\{x_i\}_{i=1}^n$, an $O(nd_s)$-time operation, where $d_s$ is the average data sparsity. Power method converges exponentially fast [5]: To achieve $\varepsilon$ accuracy, it has a total runtime of $O(nd_s \log \frac{1}{\varepsilon})$. That is, power method requires multiple passes over the full dataset.

**Online (incremental) PCA** In real-world applications, datasets might become so large that even executing a full data pass is impossible. Online learning algorithms are developed under an abstraction of this setup: They assume that data come from an "endless stream" and only process one data point (or a constant sized batch) at a time. Online PCA mostly fall under two frameworks: 1. The online worst-case scenario, where the stream of data can have a non-stationary distribution [6–8]. 2. The stochastic scenario, where one has access to i.i.d. samples from an unknown but fixed distribution [5, 9–11].

In this paper, we focus on the stochastic setup: We show that a simple variant of stochastic gradient descent (SGD), which generalizes the classic Krasulina's algorithm from $k = 1$ to general $k \geq 1$, can provably solve the $k$-PCA problem in Eq. (1.1) with an exponential convergence rate. It is worth noting that stochastic PCA algorithms, unlike batch-based solvers, can be used to optimize both the population PCA objective (1.1) and its empirical counterpart (1.2).

**Oja's method and VR-PCA** While SGD-type algorithms have iteration-wise runtime independent of the data size, their convergence rate, typically linear in the number of iterations, is significantly slower than that of batch gradient descent (GD). To speed up the convergence of SGD, the seminal work of Johnson and Zhang [12] initiated a line of effort in deriving Variance-Reduced (VR) SGD by cleverly mixing the stochastic gradient updates with occasional batch gradient updates. For convex problems, VR-SGD algorithms have provable exponential convergence rate. Despite the non-convexity of $k$-PCA problem, Shamir [5, 13] augmented Oja's method [14], a popular stochastic version of power method, with the VR step, and showed both theoretically and empirically that the resulting VR-PCA algorithm achieves exponential convergence. However, since a single VR iteration requires a full-pass over the dataset, VR-PCA is no longer an online algorithm.

**Minimax lower bound** In general, the tradeoff between convergence rate and iteration-wise computational cost is unavoidable in light of the minimax information lower bound [15, 16]: Let $\Delta^n$ (see Definition 1) denote the distance between the ground-truth rank-$k$ principal subspace and the algorithm's estimated subspace after seeing $n$ samples. Vu and Lei [15, Theorem 3.1] established that *there exists data distribution (with full-rank covariance matrices)* such that the following lower bound holds:

$$\mathbb{E}\left[\Delta^n\right] \geq \Omega(\frac{\sigma^2}{n}) \ \text{ for } \sigma^2 \geq \frac{\lambda_1 \lambda_{k+1}}{(\lambda_k - \lambda_{k+1})^2} \, , \tag{1.3}$$

Here $\lambda_k$ denotes the $k$-th largest eigenvalue of the data covariance matrix. This immediately implies a $\Omega(\frac{\sigma^2}{t})$ lower bound on the convergence rate of online $k$-PCA algorithms, since for online algorithms the number of iterations $t$ equals the number of data samples $n$. Thus, sub-linear convergence rate is impossible for online $k$-PCA algorithms on general data distributions.

## 1.1 Our result: escaping minimax lower bound on intrinsically low rank data

Despite the discouraging lower bound for online $k$-PCA, note that in Eq. (1.3), $\sigma$ equals zero when the data covariance has rank less than or equal to $k$, and consequently, the lower bound becomes un-informative. Does this imply that data low-rankness can be exploited to overcome the lower bound on the convergence rate of online $k$-PCA algorithms?

Our result answers the question affirmatively: Theorem 1 suggests that on low-rank data, an online $k$-PCA algorithm, namely, Matrix Krasulina (Algorithm 1), produces estimates of the principal subspace that converges to the ground-truth in order $O(\exp(-Ct))$, where $t$ is the number of iterations (the number of samples seen) and $C$ is a constant. Our key insight is that Krasulina's method [1], in contrast to its better-studied cousin Oja's method [14], is stochastic gradient descent with a self-regulated gradient for the PCA problem, and that when the data is of low-rank, the gradient variance vanishes as the algorithm's performance improves.

## 2 Preliminaries

We consider the following online stochastic learning setting: At time $t \in \mathbb{N} \setminus \{0\}$, we receive a random vector $X^t \in \mathbb{R}^d$ drawn i.i.d from an unknown centered probability distribution with a finite second moment. We denote by $X$ a generic random sample from this distribution. Our goal is to learn $W \in \mathbb{R}^{k' \times d}$ so as to optimize the objective in Eq (1.1).

**Notations**   We let $\Sigma^*$ denote the covariance matrix of $X$, $\Sigma^* := \mathbb{E}\left[XX^\top\right]$. We let $\{u_i\}_{i=1}^k$ denote the top $k$ eigenvectors of covariance matrix $\Sigma^*$, corresponding to its largest $k$ eigenvalues, $\lambda_1 \geq, \ldots, \geq \lambda_k$. Given that $\Sigma^*$ has rank $r$, we can represent it as: $\Sigma^* := \sum_{i=1}^r \lambda_i u_i u_i^\top$. We let $U^* := \sum_{i=1}^k u_i u_i^\top$. That is, $U^*$ is the orthogonal projection matrix into the subspace spanned by $\{u_i\}_{i=1}^k$. For any integer $p > 0$, we let $I_p$ denote the $p$-by-$p$ identity matrix. We denote by $\|\cdot\|_F$ the Frobenius norm, by $tr(\cdot)$ the trace operator. For two square matrices $A$ and $B$ of the same dimension, we denote by $A \succeq B$ if $A - B$ is positive semidefinite. We use curly capitalized letters such as $\mathcal{G}$ to denote events. For an event $\mathcal{G}$, we denote by $\mathbb{1}_\mathcal{G}$ its indicator random variable; that is, $\mathbb{1}_\mathcal{G} = 1$ if event $\mathcal{G}$ occurs and 0 otherwise.

**Optimizing the empirical objective**   We remark that our setup and theoretical results apply not only to the optimization of population $k$-PCA problem (1.1) in the infinite data stream scenario, but also to the empirical version (1.2): Given a finite dataset, we can simulate the stochastic optimization setup by sampling uniformly at random from it. This is, for example, the setup adopted by Shamir [13, 5].

**Assumptions**   In the analysis of our main result, we assume that $\Sigma^*$ has low rank and that the data norm is bounded almost surely; that is, there exits $b$ and $k$ such that

$$\mathbb{P}\left(\sup_X \|X\|^2 > b\right) = 0 \text{ and } rank(\Sigma^*) = k \tag{2.4}$$

### 2.1 Oja and Krasulina

In this section, we introduce two classic online algorithms for 1-PCA, Oja's method and Krasulina's method.

**Oja's method**   Let $w^t \in \mathbb{R}^d$ denote the algorithm's estimate of the top eigenvector of $\Sigma^*$ at time $t$. Then letting $\eta^t$ denote learning rate, and $X$ be a random sample, Oja's algorithm has the following update rule:

$$w^t \leftarrow w^{t-1} + \eta^t(XX^\top w^{t-1}) \text{ and } w^t \leftarrow \frac{w^t}{\|w^t\|}$$

We see that Oja's method is a stochastic approximation algorithm to power method. For $k > 1$, Oja's method can be generalized straightforwardly, by replacing $w^t$ with matrix $W^t \in \mathbb{R}^{k' \times d}$, and by replacing the normalization step with row orthonormalization, for example, by QR factorizaiton.

**Krasulina's method**  Krasulina's update rule is similar to Oja's update but has an additional term:

$$w^t \leftarrow w^{t-1} + \eta^t (XX^\top w^{t-1} - w^{t-1}(X^\top \frac{w^{t-1}}{\|w^{t-1}\|})^2))$$

In fact, this is stochastic gradient descent on the objective function below, which is equivalent to Eq (1.1):

$$\mathbb{E} \|X - \frac{w^t(w^t)^\top}{\|w^t\|^2} X\|^2$$

### 2.2   Gradient variance in Krasulina's method

Our key observation of Krasulina's method is as follows: Let $\tilde{w}^t := \frac{w^t}{\|w^t\|}$; Krasulina's update can be re-written as

$$w^t \leftarrow w^{t-1} + \|w^t\|\eta^t(XX^\top \tilde{w}^{t-1} - \tilde{w}^{t-1}(X^\top \tilde{w}^{t-1})^2)$$

Let

$$s^t := (\tilde{w}^t)^\top X \ \ \text{(projection coefficient)}$$

and

$$r^t := X^\top - s^t(\tilde{w}^t)^\top = X^\top - (\tilde{w}^t)^\top X(\tilde{w}^t)^\top \ \ \text{(projection residual)}$$

Krasulina's algorithm can be further written as:

$$w^t \leftarrow w^{t-1} + \|w^t\|\eta^t s^{t-1}(r^{t-1})^\top$$

The variance of the stochastic gradient term can be upper bounded as:

$$\|w^t\|^2 \operatorname{Var}\left(s^{t-1}(r^{t-1})^\top\right) \leq \|w^t\|^2 \sup_X \|X\|^2 \mathbb{E} \|r^t\|^2$$

Note that

$$\mathbb{E} \|r^t\|^2 = \mathbb{E} \|X - \frac{w^t(w^t)^\top}{\|w^t\|^2} X\|^2$$

This reveals that the variance of the gradient naturally decays as Krasulina's method decreases the $k$-PCA optimization objective. Intuitively, as the algorithm's estimated (one-dimensional) subspace $w^t$ gets closer to the ground-truth subspace $u_1$, $(w^t)^\top X$ will capture more and more of $X$'s variance, and $\mathbb{E} \|r^t\|^2$ eventually vanishes.

In our analysis, we take advantage of this observation to prove the exponential convergence rate of Krasulina's method on low rank data.

## 3   Related Works

**Stochastic optimization for PCA**  Theoretical guarantees of stochastic optimization traditionally require convexity [17]. However, many modern machine learning problems, especially those arising from deep learning and unsupervised learning, are non-convex; PCA is one of them: The objective in (1.1) is non-convex in $W$. Despite this, a series of recent theoretical works have proven stochastic optimization to be effective for PCA, mostly variants of Oja's method [18, 19, 13, 5, 20, 21, 9]. Krasulina's method [1] was much less studied than Oja's method; a notable exception is the work of Balsubramani et al. [9], which proved $O(1/t)$ rate in expectation for both Oja's and Krasulina's algorithm for 1-PCA. We also noticed a recent pre-print [22] that analyzes Krasulina's algorithm, which establishes $O(1/\sqrt{t})$ convergence with high probability.

**Stochastic optimization for $k$-PCA**  There were very few theoretical analysis of stochastic $k$-PCA algorithms with $k > 1$, with the exception of Allen-Zhu and Li [18], Shamir [13], Balcan et al. [23], Li et al. [24]. All had focused on variants of Oja's algorithm, among which Shamir [13] was the only previous work, to the best of our knowledge, that provided a local exponential convergence rate guarantee of Oja's algorithm for $k \geq 1$. Their result holds for general data distribution, but their variant of Oja's algorithm, VR-PCA, requires several full passes over the datasets, and thus not fully online.

---

**Algorithm 1** Matrix Krasulina

---

**Input:** Initial matrix $W^o \in \mathbb{R}^{k' \times d}$; learning rate schedule $(\eta^t)$; number of iterations, $T$;
**while** $t \leq T$ **do**
    1. Sample $X^t$ i.i.d. from the data distribution

    2. Orthonormalize the rows of $W^{t-1}$ (e.g., via QR factorization)

    3. $W^t \leftarrow W^{t-1} + \eta^t W^{t-1} X^t (X^t - (W^{t-1})^\top W^{t-1} X^t)^\top$
**end while**
**Output:** $W^\top$

---

The machine learning and computer science community has studied the PCA problem without imposing strong assumptions on data. A typical assumption would be a gap on the eigenvalues [9, 21, 13, 24, 23]; recent work of Allen-Zhu and Li [18] has even removed this assumption. The subspace tracking literature has approached the problem from a different angle (see Balzano et al. [25] for an overview).

**Subspace tracking**    Under a generative model assumption, $X = \bar{U}s$, where $\bar{U} \in \mathbb{R}^{d \times k}$ is a basis of a $k$-dimensional subspace and $Cov(s) = I_k$ [26, Condition 1], Zhang and Balzano [26] established the global convergence (in a different subspace distance metric than ours) of a subspace tracking algorithm called GROUSE [27]. However, no theoretical guarantee on GROUSE was provided without the generative model assumption, which also implies data low-rankness.

Our Theorem 1 shows that when the data is of low-rank Matrix Krasulina can achieve local exponential convergence, and Theorem 2 shows that without low-rank assumption Matrix Krasulina has $O(\frac{1}{t})$ local convergence rate. Our Theorem 3 provides preliminary results on how to make the convergence global.

## 4   Main results

Generalizing vector $w^t \in \mathbb{R}^d$ to matrix $W^t \in \mathbb{R}^{k' \times d}$ as the algorithm's estimate at time $t$, we derive *Matrix Krasulina* (Algorithm 1), so that the row space of $W^t$ converges to the $k$-dimensional subspace spanned by $\{u_1, \ldots, u_k\}$.

### 4.1   Matrix Krasulina

Inspired by the original Krasulina's method, we design the following update rule for the Matrix Krasulina (Algorithm 1): Let
$$s^t := W^{t-1} X^t \quad \text{and} \quad r^t := X^t - (W^{t-1})^\top (W^{t-1}(W^{t-1})^\top)^{-1} W^{t-1} X^t \,,$$
Since we impose an orthonormalization step in Algorithm 1, $r^t$ is simplified to
$$r^t := X^t - (W^{t-1})^\top W^{t-1} X^t \,,$$
Then the update rule of Matrix Krasulina can be re-written as
$$W^t \leftarrow W^{t-1} + \eta^t s^t (r^t)^\top \,,$$
For $k' = 1$, this reduces to Krasulina's update with $\|w^t\| = 1$. The self-regulating variance argument for the original Krasulina's method still holds, that is, we have
$$\mathbb{E} \|s^t (r^t)^\top\|^2 \leq b \, \mathbb{E} \|r^t\|^2 = b \, \mathbb{E} \|X - (W^t)^\top W^t X\|^2 \,,$$
where $b$ is as defined in Eq (2.4). We see that the last term coincides with the objective in Eq. (1.1).

#### 4.1.1   Loss measure

Given the algorithm's estimate $W^t$ at time $t$, we let $P^t$ denote the orthogonal projection matrix into the subspace spanned by its rows, $\{W^t_{i,\star}\}_{i=1}^{k'}$, that is,
$$P^t := (W^t)^\top (W^t (W^t)^\top)^{-1} W^t = (W^t)^\top W^t \,,$$
In our analysis, we use the following loss measure to track the evolvement of $W^t$:

**Definition 1** (Subspace distance). *Let $\mathcal{S}$ and $\hat{\mathcal{S}}^t$ be the ground-truth principal subspace and its estimate of Algorithm 1 at time $t$ with orthogonal projectors $U^*$ and $P^t$, respectively. We define the subspace distance between $S$ and $\hat{\mathcal{S}}^t$ as $\Delta^t := tr(U^*(I - P^t)) = k - tr(U^*P^t)$.*

Note that $\Delta^t$ in fact equals the sum of squared canonical angles between $\mathcal{S}$ and $\hat{\mathcal{S}}^t$, and coincides with the subspace distance measure used in related theoretical analyses of $k$-PCA algorithms [18, 13, 15].

### 4.2 Convergence rates of Matrix Krasulina

This section presents our convergence results; proofs are deferred to the Appendix. Our first theorem proves the exponential convergence rate of Matrix Krasulina measured by $\Delta^t$.

**Theorem 1** (Exponential convergence with constant learning rate). *Suppose assumption Eq. (2.4) holds. Suppose the initial estimate $W^o \in \mathbb{R}^{k' \times d}$ ($k' \geq k$) in Algorithm 1 satisfies that, for some $\tau \in (0, 1)$,*

$$\Delta^o \leq 1 - \tau \,,$$

*Suppose for any $\delta > 0$, we choose a constant learning rate $\eta^t = \eta$ such that*

$$\eta \leq \min\left\{ \frac{\sqrt{2}-1}{b}, \frac{\lambda_k \tau}{\lambda_1 b(k+3)}, \frac{2\lambda_k \tau}{\frac{8}{1-\tau}\ln\frac{1}{\delta}(b + \|\Sigma^*\|_F)^2 + b(k+1)\lambda_1} \right\} \,,$$

*Then there exists event $\mathcal{G}_t$ such that $\mathbb{P}(\mathcal{G}_t) \geq 1 - \delta$, and*

$$\mathbb{E}\left[\Delta^t | \mathcal{G}_t\right] \leq \frac{1}{1 - \delta} \exp\left(-t\eta\tau\lambda_k\right) \,.$$

On a high level, Theorem 1 is proved in the following steps (all proofs are deferred to the Appendix):

**In section A.2** We show that if the algorithm's iterates, $W^t$, stay inside the basin of attraction, which we formally define as event $\mathcal{G}_t$, $\mathcal{G}_t := \left\{\Delta^i \leq 1 - \tau, \forall i \leq t\right\}$, then a suitable transformation of the stochastic process $(\Delta^t)$ forms a supermartingale.

**In section A.3** Using martingale concentration inequality, we show that provided a good initialization, it is likely that *the algorithm's outputs $W^1, \ldots, W^t$ stay inside the basin of attraction*.

**In section A.4** We show that at each iteration $t$, conditioning on $\mathcal{G}_t$, $\Delta^{t+1} \leq \beta\Delta^t$ for some $\beta < 1$ if we set the learning rate $\eta^t$ to be a properly chosen constant.

**In section D** We iteratively apply this recurrence relation to prove Theorem 1.

From Theorem 1, we observe that (a). The convergence rate of Algorithm 1 on strictly low-rank data does not depend on the data dimension $d$, but only on the intrinsic dimension $k$. This is verified by our experiments (see Sec. 5). (b). We see that the learning rate should be of order $O(\frac{\lambda_k}{k\lambda_1})$: Empirically, we found that setting $\eta$ to be roughly $\frac{1}{10\lambda_1}$ gives us the best convergence result. Note, however, this learning rate setup is not practical since it requires knowledge of eigenvalues.

**Comparison between Theorem 1 and Shamir [13, Theorem 1]** (1). The result in Shamir [13] does not rely on the low-rank assumption of $\Sigma^*$. Since the variance of update in Oja's method is not naturally decaying, they use VR technique inspired by Johnson and Zhang [12] to reduce the variance of the algorithm's iterate, which is computationally heavy: the block version of VR-PCA converges at rate $O(\exp(-CT))$, where $T$ denotes the number of data passes. (2). Our result has a similar learning rate dependence on the data norm bound $b$ as that of Shamir [13, Theorem 1]. (3). The initialization requirement in Theorem 1 is comparable to Shamir [13, Theorem 1]. (4). Conditioning on the event of successful convergence, their exponential convergence rate result holds deterministically, whereas our convergence rate guarantee holds in expectation.

While our main focus is on taking advantage of low-rank data, the next theorem shows that on full-rank datasets, if we tune the learning rate to decay at order $O(\frac{1}{t})$, then the algorithm achieves $O(\frac{1}{t})$ convergence.

---
**Algorithm 2** Warm-start with Matrix Krasulina
---
**Input:** Epoch budget $N$; inner loop budget $T$, learning rate $\eta$; number of rows in initial matrix $k'$; shrinkage factor $\rho$;
Set $k_o \leftarrow k'$ and initialize $W_o^o \in \mathbb{R}^{k_o \times d}$ with entries $(W_o^o)_{ij} \sim \mathcal{N}(0,1)$.
**while** $i < N$ **do**
    **while** $t < T$ **do**
        Update $W_i^{t+1} \leftarrow W_i^t$ by running Matrix Krasulina iteration with learning rate schedule $\eta$
    **end while**
    Set $k_{i+1} \leftarrow k_i(1 - \rho)$, and construct $W_{i+1}^o$ by randomly sampling $k_{i+1}$ rows from $W_i^{T-1}$.
**end while**
---

**Theorem 2** (Linear convergence on full rank data). *Suppose* $\mathbb{P}\left(\sup \|X\|^2 > b\right) = 0$. *Suppose the initial estimate* $W^o \in \mathbb{R}^{k \times d}$ *in Algorithm 1 satisfies* $\Delta^o \leq \frac{1-\tau}{2}$, *for some* $\tau \in (0,1)$. *Let the learning rate schedule be* $\eta^t = \frac{c}{t_o+t}$, *for some constants* $c, t_o$, *and let* $B := \max\left(8(b + \|\Sigma^*\|_F)^2 k, (kb + 2cb^2 + c^2b^3)\lambda_1(d-k)\right)$. *If we choose* $c, t_o$ *such that*

$$c \geq \frac{1}{(\lambda_k - \lambda_{k+1})\tau} \quad and \quad t_o \geq \max\{\frac{64Bc^2 \ln \frac{1}{\delta}}{(\Delta^o)^2}, 1\},$$

*Then for any* $\delta \in (0, \frac{1}{e})$, *there exists event* $\mathcal{G}_t$ *such that* $\mathbb{P}(\mathcal{G}_t) \geq 1 - \delta$, *and* $\mathbb{E}[\Delta^t|\mathcal{G}_t] \leq O(\frac{1}{t})$.

Theorem 2 generalizes the result of [9], where linear convergence rate of Krasulina's algorithm is established for the 1-PCA problem on full-rank data. The linear convergence rate on full-rank data matches that of the minimax lower bound in Eq (1.3) up to constants (note that here the initialization condition is more strict than Theorem 1; whether this is an artifact of our analysis is left to future work).

## 4.3 Random initialization guarantee of $W^o$

Theorem 1 focuses on the convergence rate of Matrix Krasulina from a good initialization point. Next, we show that if we are willing to use $k' > k$ rows in $W^o$, then randomly initializing the weights in $W^o$ is sufficient to guarantee the initialization requirement, $\Delta^o \leq 1 - \tau$.

**Theorem 3** (Success guarantee of an over-complete initialization). *Let* $\varepsilon, t > 0$ *be any constants. If we choose* $k' \geq \frac{1+t}{1-\varepsilon}(1 - \frac{1-\tau}{k})d$, *then with probability at least* $1 - 2k \exp\left(-(\varepsilon^2 - \varepsilon^3)k'/4\right) - \frac{k'+1}{dt^2}$,

$$\Delta^o \leq 1 - \tau.$$

The proof is a simple application of Lemma 4 in Section F.

**How large should $k'$ be given $d, k$?** In the special case of $k = 1$, we can choose $\tau = \frac{1}{d}\frac{1-\varepsilon}{1+t}$, and then we get the lower bound $k' \geq 1$. We need to choose a larger $k'$ as the intrinsic rank $k$ gets larger. In general, $k'$ is of order $\Omega(d - \frac{d}{k})$.

**A phase-wise warm start with over-complete random initialization** As seen from previous discussion, using vanilla random initialization is only reasonable if the ratio $\frac{k}{d}$ is small. Otherwise, the number of rows in $W^o$, $k'$, can almost be as large as $d$. To deal with this drawback, inspired by Oja++ of Allen-Zhu and Li [18], we propose a warm-start strategy as Algorithm 2. The main insight is captured by the following lemma:

**Lemma 1.** *For any* $i > 0$, *at the end of $i$-th epoch of Algorithm 2*,

$$\mathbb{E}\left[tr(U^*P(W_i^o))\right] \geq \frac{k_i}{k_{i-1}} \mathbb{E}\left[tr(U^*P(W_{i-1}^{T-1}))\right]$$

Note that the error of the first iterate at the $i$-th epoch is $\Delta(W_i^o) = k - tr(U^*P(W_i^o))$. So Lemma 1 quantifies how much the error is increased between the last iterate of epoch $i - 1$ and the first iterate of epoch $i$, due to the row-sampling step at the end of $i-1$-th epoch. Based on Lemma 1, the intuition

of why Algorithm 2 works is as follows: At the initial epoch, we choose $k_o$ to be large enough to satisfy the condition of Theorem 3. Then Theorem 1 implies that after $T$ inner loop iterations, the expected error will decrease in order $O(-\eta\tau\lambda_k T)$; now if we choose a suitable shrinkage factor $\rho$, then we can guarantee that the error of $W_1^o$, although larger than $W_o^{T-1}$, will still satisfy the condition in Theorem 3. Thus, applying Theorem 1, we can decrease the error of $W_1^o$ rapidly again, and so on. Eventually, after $O(\log \frac{k_o}{k})$ epochs, we will obtain a matrix of $O(k)$ number of rows, while satisfying the condition of Theorem 3. We leave the formal analysis and empirical evaluation of Algorithm 2 to future work.

### 4.4 Open question: extending our result to effectively low-rank data

Many real-world datasets are not strictly low-rank, but *effectively low-rank* (see, for example, Figure 2): Informally, we say a dataset is effectively low-rank **if there exists $k \ll d$ such that** $\frac{\sum_{i>k}\lambda_i}{\sum_{j\le k}\lambda_j}$ **is small** , We conjecture that our analysis can be adapted to show theoretical guarantee of Algorithm 1 on effectively low-rank datasets as well. In Section 5, our empirical results support this conjecture. Formally characterizing the dependence of convergence rate on the "effective low-rankness" of a dataset can provide a smooth transition between the linear convergence lower bound [15] and our result in Theorem 1.

## 5   Experiments

In this section, we present our empirical evaluation of Algorithm 1 to understand its convergence property on low-rank or effectively low-rank datasets [2]. We first verified its performance on simulated low-rank data and effectively low-rank data, and then we evaluated its performance on two real-world effectively low-rank datasets.

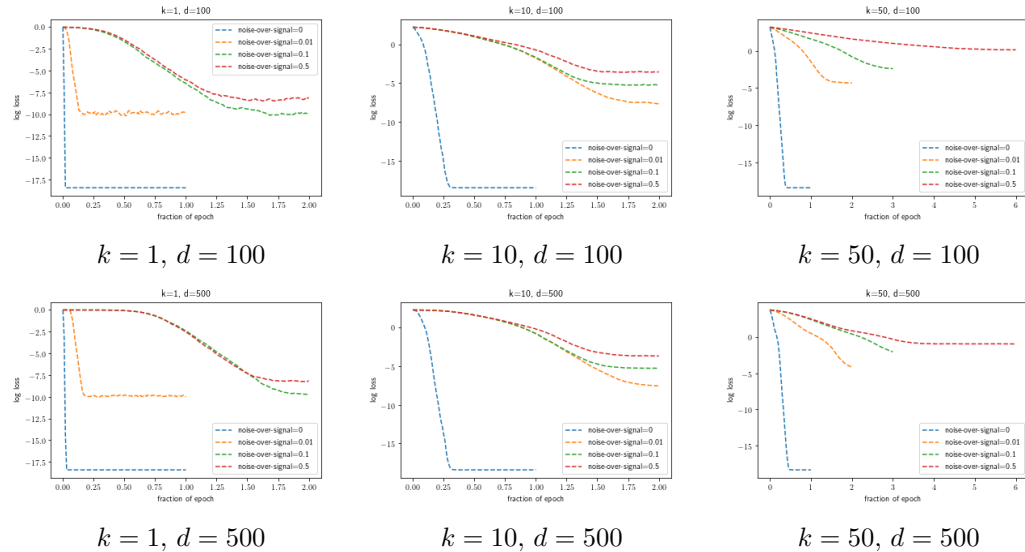

Figure 1: log-convergence graph of Algorithm 1: $\ln(\Delta^t)$ vs $t$ at different levels of noise-over-signal ratio ($\frac{\sum_{i>k}\lambda_i}{\sum_{j\le k}\lambda_j}$)

### 5.1   Simulations

The low-rank data is generated as follows: we sample i.i.d. standard normal on the first $k$ coordinates of the $d$-dimensional data (the rest $d-k$ coordinates are zero), then we rotate all data using a random orthogonal matrix (unknown to the algorithm).

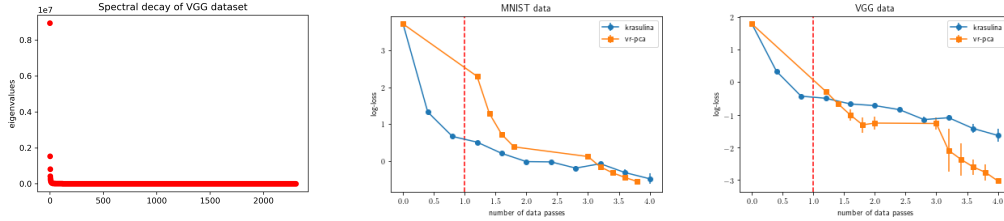

Figure 2: top 6 eigenvalues explains 80% of the data variance.     MNIST ($d = 784; k' = 44$)     VGG ($d = 2304; k' = 6$); red vertical line marks a full pass over the dataset

**Simulating effectively low-rank data**    In practice, hardly any dataset is strictly low-rank but many datasets have sharply decaying spectra (recall Figure 2). Although our Theorem 1 is developed under a strict low-rankness assumption, here we empirically test the robustness of our convergence result when data is not strictly low rank but only effectively low rank. Let $\lambda_1, \geq \cdots \geq \lambda_d \geq 0$ be the spectrum of a covariance matrix. For a fixed $k \in [d]$, we let *noise-over-signal* $:= \frac{\sum_{i>k} \lambda_i}{\sum_{j \leq k} \lambda_j}$ . The *noise-over-signal* ratio intuitively measures how "close" the matrix is to a rank-$k$ matrix: The smaller the number is, the shaper the spectral decay; when the ratio equals zero, the matrix is of rank at most $k$. In our simulated data, we perturb the spectrum of a strictly rank-$k$ covariance matrix and generate data with full-rank covariance matrices at the following noise-over-signal ratios, $\{0, 0.01, 0.1, 0.5\}$.

**Results**    Figure 1 shows the *log-convergence graph* of Algorithm 1 on our simulated data: We initialized Algorithm 1 with a random matrix $W^o$ and ran it for one or a few epochs, each consists of 5000 iterations. (1). We verified that, on strictly low rank data (noise-over-signal= 0), the algorithm indeed has an exponentially convergence rate (linear in log-error); (2). As we increase the noise-over-signal ratio, the convergence rate gradually becomes slower; (3). The convergence rate is not affected by the actual data dimension $d$, but only by the intrinsic dimension $k$, as predicted by Theorem 1.

## 5.2    Real effectively low-rank datasets

We take a step further to test the performance of Algorithm 1 on two real-world datasets: VGG [28] is a dataset of 10806 image files from 2622 distinct celebrities crawled from the web, with $d = 2304$. For MNIST [29], we use the 60000 training examples of digit pixel images, with $d = 784$. Both datasets are full-rank, but we choose $k'$ such that the noise-over-signal ratio at $k'$ is 0.25; that is, the top $k'$ eigenvalues explain 80% of data variance. We compare Algorithm 1 against the exponentially convergent VR-PCA: we initialize the algorithms with the same random matrix and we train (and repeated for 5 times) using the best constant learning rate we found empirically for each algorithm. We see that Algorithm 1 retains fast convergence even if the datasets are not strictly low rank, and that it has a clear advantage over VR-PCA before the iteration reaches a full pass; indeed, VR-PCA requires a full-pass over the dataset before its first iterate.

**Acknowledgments**

Cheng would like to thank all anonymous reviewers and the meta-reviewer for providing insightful feedbacks on improving the quality of this paper. Cheng is very grateful to her PhD advisor, Claire Monteleoni, for her kind encouragement in pursuing this project, and to Amazon Web Services for various supports.

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

## Footnotes

*A major part of this work was done prior to the author joining Amazon when she was a student at George Washington University.

[2]Code will be available at `https://github.com/chengtang48/neurips19`.
