[Supplementary Material]

# A   Sketch of analysis for Theorem 1

**Additional notations:**   Before proceeding to our analysis, we introduce some technical notations for stochastic processes: Let $(\mathcal{F}_t)$ denote the natural filtration (collection of $\sigma$-algebras) associated to the stochastic process, that is, the data stream $(X^t)$. Then by the update rule of Algorithm 1, for any $t$, $W^t$, $P^t$, and $\Delta^t$ are all $\mathcal{F}_t$-measurable, and $\mathcal{G}_t \in \mathcal{F}_t$.

## A.1   A roadmap of analysis

On a high level, proving Theorem 1 is done in the following steps:

**In section A.2**   We show that if the algorithm's iterates, $W^t$, stay inside the basin of attraction, which we formally define as event $\mathcal{G}_t$, $\mathcal{G}_t := \{\Delta^i \leq 1 - \tau, \forall i \leq t\}$, then a suitable transformation of the stochastic process $(\Delta^t)$ forms a supermartingale.

**In section A.3**   Using martingale concentration inequality, we show that provided a good initialization, it is likely that *the algorithm's outputs $W^1, \ldots, W^t$ stay inside the basin of attraction*.

**In section A.4**   We show that at each iteration $t$, conditioning on $\mathcal{G}_t$, $\Delta^{t+1} \leq \beta\Delta^t$ for some $\beta < 1$ if we set the learning rate $\eta^t$ to be a properly chosen constant.

**In section D**   We iteratively apply this recurrence relation to prove Theorem 1.

## A.2   A conditional supermartingale

Letting $M_i := \mathbb{1}_{\mathcal{G}_{i-1}} \exp\left(s\Delta^i\right)$, Lemma 1 shows that $(M_i)_{i \geq 1}$ forms a supermartingale.

**Lemma 1** (Supermartingale construction)**.** *Suppose $\mathcal{G}_0$ holds. Let $C^t$ and $Z$ be as defined in Proposition 2. Then for any $i \leq t$, and for any constant $s > 0$,*

$$\mathbb{E}\left[\mathbb{1}_{\mathcal{G}_i} \exp\left(s\Delta^{i+1}\right)|\mathcal{F}_i\right]$$
$$\leq \mathbb{1}_{\mathcal{G}_{i-1}} \exp\left(s\Delta^i\left(1 - 2\eta^{i+1}\lambda_k\tau + (\eta^{i+1})^2 C^{i+1}\lambda_1\right) + 2s^2(\eta^{i+1})^2|Z|^2\right).$$

The proof of Lemma 1 utilizes the iteration-wise convergence inequality in Prop. 2 of Section A.4.

## A.3   Bounding probability of bad event $\mathcal{G}_t^c$

Let $\mathcal{G}_0$ denote the good event happening upon initialization of Algorithm 1. Observe that the good events form a nested sequence of subsets through time:

$$\mathcal{G}_0 \supset \mathcal{G}_1 \supset \ldots \mathcal{G}_t \supset \ldots$$

This implies that we can partition the bad event $\mathcal{G}_t^c$ into a union of individual bad events:

$$\mathcal{G}_t^c = \cup_{i=1}^t \left( \mathcal{G}_{i-1} \setminus \mathcal{G}_i \right),$$

The idea behind Proposition 1 is that, we first transform the union of events above into a maximal inequality over a suitable sequence of random variables, which form a supermartingale, and then we apply a type of martingale large-deviation inequality to upper bound $\mathbb{P}\left(\mathcal{G}_t^c\right)$.

**Proposition 1** (Bounding probability of bad event). *Suppose the initialization condition in Theorem 1 holds. For any $\delta > 0$, $t \geq 1$, and $i \leq t$, if the learning rate $\eta^i$ is set such that*

$$\eta^i \leq \min \left\{ \frac{2\lambda_k \tau}{\left( \frac{16}{1-\tau} \ln \frac{1}{\delta} (b + \|\Sigma^*\|_F)^2 + b(k+1)\lambda_1 \right)}, \frac{\sqrt{2}-1}{b} \right\},$$

*Then $\mathbb{P}\left(\mathcal{G}_t^c\right) \leq \delta$.*

*Proof Sketch.* For $i > 1$, we first consider the individual events:

$$\mathcal{G}_{i-1} \setminus \mathcal{G}_i = \mathcal{G}_{i-1} \cap \mathcal{G}_i^c = \{\forall j < i, \ \Delta^j \leq 1 - \tau\} \cap \{\Delta^i > 1 - \tau\}$$

For any strictly increasing positive measurable function $g$, the above is equivalent to

$$\mathcal{G}_{i-1} \setminus \mathcal{G}_i = \{g(\Delta^i) > g(1-\tau) \ \text{and} \ \forall j < i, \ g(\Delta^j) \leq g(1-\tau)\}$$

Since event $\mathcal{G}_{i-1}$ occurs is equivalent to $\{\mathbb{1}_{\mathcal{G}_{i-1}} = 1\}$, we can write

$$\mathcal{G}_{i-1} \setminus \mathcal{G}_i = \{g(\Delta^i) > g(1-\tau) \ \text{and} \ \forall j < i, \ g(\Delta^j) \leq g(1-\tau), \ \text{and} \ \mathbb{1}_{\mathcal{G}_{i-1}} = 1\}$$

Additionally, since for any $j' < j$, $\mathcal{G}_{j'} \supset \mathcal{G}_j$, that is, $\{\mathbb{1}_{\mathcal{G}_j} = 1\}$ implies $\{\mathbb{1}_{\mathcal{G}_{j'}} = 1\}$, we have

$$\mathcal{G}_{i-1} \setminus \mathcal{G}_i$$
$$= \{g(\Delta^i) > g(1-\tau) \ \text{and} \ \forall j < i, \ g(\Delta^j) \leq g(1-\tau), \mathbb{1}_{\mathcal{G}_{i-1}} = 1, \mathbb{1}_{\mathcal{G}_{j'}} = 1, \forall j' < i - 1\}$$
$$= \{\mathbb{1}_{\mathcal{G}_{i-1}} g(\Delta^i) > g(1-\tau) \ \text{and} \ \forall j < i, \mathbb{1}_{\mathcal{G}_{j-1}} g(\Delta^j) \leq g(1-\tau), \ \text{and} \ \mathbb{1}_{\mathcal{G}_j} = 1\}$$
$$\subset \{\mathbb{1}_{\mathcal{G}_{i-1}} g(\Delta^i) > g(1-\tau) \ \text{and} \ \forall j < i, \mathbb{1}_{\mathcal{G}_{j-1}} g(\Delta^j) \leq g(1-\tau)\}$$

So the union of the terms $\mathcal{G}_{i-1} \setminus \mathcal{G}_i$ can be upper bounded as

$$\cup_{i=1}^t \mathcal{G}_{i-1} \setminus \mathcal{G}_i \subset$$
$$\cup_{i=2}^t \{\mathbb{1}_{\mathcal{G}_{i-1}} g(\Delta^i) > g(1-\tau), \mathbb{1}_{\mathcal{G}_{j-1}} g(\Delta^j) \leq g(1-\tau), \forall 1 \leq j < i\}$$
$$\cup \{\mathbb{1}_{\mathcal{G}_0} g(\Delta^1) > g(1-\tau)\}$$

Observe that the event above can also be written as

$$\{ \sup_{1 \leq i \leq t} \mathbb{1}_{\mathcal{G}_{i-1}} g(\Delta^i) > g(1-\tau)\}.$$

We upper bound the probability of the event above by applying a variant of Doob's inequality. To achieve this, the key step is to find a suitable function $g$ such that the sequence

$$\mathbb{1}_{\mathcal{G}_0}\, g(\Delta^1), \mathbb{1}_{\mathcal{G}_1}\, g(\Delta^2), \ldots, \mathbb{1}_{\mathcal{G}_{i-1}}\, g(\Delta^i), \ldots$$

forms a supermartingale. Via Lemma 1, we show that if we choose $g(x) := \exp(sx)$ for any constant $s > 0$, then

$$\mathbb{E}\left[\mathbb{1}_{\mathcal{G}_i} \exp\left(s\Delta^{i+1}\right) | \mathcal{F}_i\right] \leq \mathbb{1}_{\mathcal{G}_{i-1}} \exp\left(s\Delta^i\right), \tag{A.1}$$

provided we choose the learning rate in Algorithm 1 appropriately. Then a version of Doob's inequality for supermartingale (2, p. 231) implies that

$$\mathbb{P}\left(\sup_i \mathbb{1}_{\mathcal{G}_{i-1}} \exp\left(s\Delta^i\right) > \exp\left(s(1-\tau)\right)\right) \leq \frac{\mathbb{E}\left[\mathbb{1}_{\mathcal{G}_0} \exp\left(s\Delta^1\right)\right]}{\exp\left(s(1-\tau)\right)},$$

Finally, bounding the expectation on the RHS using our assumption on the initialization condition finishes the proof. □

## A.4  Iteration-wise convergence result

**Proposition 2** (Iteration-wise subspace improvement). *At the $t+1$-th iteration of Algorithm 1, the following holds:*

*(V1) Let $C^t := kb + 2\eta^t b^2 + (\eta^t)^2 b^3$. Then*

$$\mathbb{E}\left[tr(U^* P^{t+1}) | \mathcal{F}_t\right] \geq tr(U^* P^t) + 2\eta^{t+1}\lambda_k \Delta^t (1 - \Delta^t) - (\eta^{t+1})^2 C^{t+1} \lambda_1 \Delta^t$$

*(V2) There exists a random variable $Z$, with*

$$\mathbb{E}\left[Z | \mathcal{F}_t\right] = 0 \text{ and } |Z| \leq 2(b + \|\Sigma^*\|_F)\sqrt{\Delta^t}$$

*such that*

$$tr(U^* P^{t+1}) \geq tr(U^* P^t) + 2\eta^{t+1}\lambda_k \Delta^t (1 - \Delta^t) + 2\eta^{t+1} Z - (\eta^{t+1})^2 C^{t+1} \lambda_1 \Delta^t$$

*Proof Sketch.* By definition,

$$tr(U^* P^{t+1}) = tr(U^* (W^{t+1})^\top (W^{t+1}(W^{t+1})^\top)^{-1} W^{t+1}),$$

where by the update rule of Algorithm 1

$$W^{t+1} = W^t + \eta^{t+1} s^{t+1} (r^{t+1})^\top.$$

We first derive (V1); the proof sketch is as follows:

1. Since the rows of $W^t$ are orthonormalized, one would expect that a small perturbation of this matrix, $W^{t+1}$, is also close to orthonormalized, and thus $W^{t+1}(W^{t+1})^\top$ should be close to an identity matrix. Lemma 2 shows this is indeed the case, offsetting by a small term $E$, which can be viewed as an error/excessive term:

**Lemma 2** (Inverse matrix approximation). *Let $k'$ be the number of rows in $W^t$. Suppose the rows of $W^t$ are orthonormal, that is, $W^t(W^t)^\top = I_{k'}$ . Then for $W^{t+1} = W^t + \eta^{t+1}s^{t+1}(r^{t+1})^\top$ , we have*

$$(W^{t+1}(W^{t+1})^\top)^{-1} \succeq (1 - \lambda_1(E))I_{k'} ,$$

*where $\lambda_1(E)$ is the largest eigenvalue of some matrix E, and $\lambda_1(E) = (\eta^{t+1})^2\|r^{t+1}\|^2\|s^{t+1}\|^2$ .*

This implies that

$$tr U^*(W^{t+1})^\top(W^{t+1}(W^{t+1})^\top)^{-1}W^{t+1}$$
$$\geq (1 - (\eta^{t+1})^2\|r^{t+1}\|^2\|s^{t+1}\|^2)tr(U^*(W^{t+1})^\top W^{t+1})$$

2. We continue to lower bound the conditional expectation of the last term in the previous inequality as

$$\mathbb{E}\left[tr(U^*(W^{t+1})^\top W^{t+1})|\mathcal{F}_t\right] \geq tr(U^*P^t) + 2\eta^{t+1}tr(U^*P^t\Sigma^*(I_d - P^t))$$

3. The last term in the inequality above, $tr(U^*P^t\Sigma^*(I_d - P^t))$, controls the improvement in proximity between the estimated and the ground-truth subspaces. In Lemma 3, we lower bound it as a function of $\Delta^t$:

   **Lemma 3** (Characterization of stationary points). *Let $\Sigma^*$ be of rank k, and*

   $$\Gamma^t := tr(U^*P^t\Sigma^*(I_d - P^t)) ,$$

   *Then the following holds:*

   *(a) $tr(U^*P^t) = tr(U^*)$ implies that $\Gamma^t = 0$ .*
   *(b) $\Gamma^t \geq \lambda_k\Delta^t(1 - \Delta^t)$ .*

4. Finally, combining the results above, we obtain (V1) inequality in the statement of the proposition.

(V2) inequality is derived similarly with the steps above, except that at step 2, instead of considering the conditional expectation of $tr(U^*(W^{t+1})^\top W^{t+1})$, we explicitly represent the zero-mean random variable $Z$ in the inequality. $\square$

# B  Proofs for Proposition 1

*Proof of Proposition 1.* Recall definition of $\mathcal{G}_t$, $\mathcal{G}_t := \{\Delta^i \leq 1 - \tau, \forall i \leq t\}$ . We partition its complement as $\mathcal{G}_t^c = \cup_{i=1}^t \mathcal{G}_{i-1} \setminus \mathcal{G}_i$ . For $i > 1$, we first consider the individual events:

$$\mathcal{G}_{i-1} \setminus \mathcal{G}_i = \mathcal{G}_{i-1} \cap \mathcal{G}_i^c = \{\Delta^i > 1 - \tau\} \cap \{\forall j < i, \, \Delta^j \leq 1 - \tau\}$$

For any strictly increasing positive measurable function $g$, the above is equivalent to

$$\mathcal{G}_{i-1} \setminus \mathcal{G}_i = \{g(\Delta^i) > g(1 - \tau) \text{ and } \forall j < i, \, g(\Delta^j) \leq g(1 - \tau)\}$$

Since event $\mathcal{G}_{i-1}$ occurs is equivalent to $\{\mathbb{1}_{\mathcal{G}_{i-1}} = 1\}$, we can write

$$\mathcal{G}_{i-1} \setminus \mathcal{G}_i = \{g(\Delta^i) > g(1-\tau) \text{ and } \forall j < i, \ g(\Delta^j) \le g(1-\tau), \text{ and } \mathbb{1}_{\mathcal{G}_{i-1}} = 1\}$$

Additionally, since for any $j' < j$, $\mathcal{G}_{j'} \supset \mathcal{G}_j$, that is, $\{\mathbb{1}_{\mathcal{G}_j} = 1\}$ implies $\{\mathbb{1}_{\mathcal{G}_{j'}} = 1\}$, we have

$$\mathcal{G}_{i-1} \setminus \mathcal{G}_i$$
$$= \{g(\Delta^i) > g(1-\tau) \text{ and } \forall j < i, \ g(\Delta^j) \le g(1-\tau), \mathbb{1}_{\mathcal{G}_{i-1}} = 1, \mathbb{1}_{\mathcal{G}_{j'}} = 1, \forall j' < i-1\}$$
$$= \{\mathbb{1}_{\mathcal{G}_{i-1}} g(\Delta^i) > g(1-\tau) \text{ and } \forall j < i, \mathbb{1}_{\mathcal{G}_{j-1}} g(\Delta^j) \le g(1-\tau), \text{ and } \mathbb{1}_{\mathcal{G}_j} = 1\}$$
$$\subset \{\mathbb{1}_{\mathcal{G}_{i-1}} g(\Delta^i) > g(1-\tau) \text{ and } \forall j < i, \mathbb{1}_{\mathcal{G}_{j-1}} g(\Delta^j) \le g(1-\tau)\}$$

So the union of the terms $\mathcal{G}_{i-1} \setminus \mathcal{G}_i$ can be upper bounded as

$$\cup_{i=1}^t \mathcal{G}_{i-1} \setminus \mathcal{G}_i \subset$$
$$\cup_{i=2}^t \{\mathbb{1}_{\mathcal{G}_{i-1}} g(\Delta^i) > g(1-\tau), \mathbb{1}_{\mathcal{G}_{j-1}} g(\Delta^j) \le g(1-\tau), \forall 1 \le j < i\}$$
$$\cup \{\mathbb{1}_{\mathcal{G}_0} g(\Delta^1) > g(1-\tau)\}$$

Observe that the event above can also be written as

$$\left\{ \sup_{1 \le i \le t} \mathbb{1}_{\mathcal{G}_{i-1}} g(\Delta^i) > g(1-\tau) \right\}.$$

Now we upper bound $\mathbb{P}\left(\{\sup_{1 \le i \le t} \mathbb{1}_{\mathcal{G}_{i-1}} g(\Delta^i) > g(1-\tau)\}\right)$ by applying a martingale large deviation inequality. To achieve this, the key step is to find a suitable function $g$ such that the stochastic process

$$\mathbb{1}_{\mathcal{G}_0} g(\Delta^1), \mathbb{1}_{\mathcal{G}_1} g(\Delta^2), \ldots, \mathbb{1}_{\mathcal{G}_{i-1}} g(\Delta^i), \ldots$$

is a supermartingale. In this proof, we choose $g : \mathbb{R} \to \mathbb{R}_{>0}$ to be $g(x) = \exp(sx)$ for $s = \frac{1}{1-\tau} \ln \frac{1}{\delta}$.
By Lemma 1,

$$\mathbb{E}\left[\mathbb{1}_{\mathcal{G}_i} \exp\left(s\Delta^{i+1}\right) | \mathcal{F}_i\right]$$
$$\le \mathbb{1}_{\mathcal{G}_{i-1}} \exp\left(s\Delta^i \left(1 - 2\eta^{i+1}\lambda_k\tau + (\eta^{i+1})^2 C^{i+1}\lambda_1\right) + 2s^2(\eta^{i+1})^2 |Z|^2\right)$$
$$\le \mathbb{1}_{\mathcal{G}_{i-1}} \exp\left(s\Delta^i \left(1 - 2\eta^{i+1}\lambda_k\tau + (\eta^{i+1})^2 C^{i+1}\lambda_1\right)\right) \exp\left(s^2(\eta^{i+1})^2 8(b + \|\Sigma^*\|_F)^2 \Delta^i\right)$$
$$= \mathbb{1}_{\mathcal{G}_{i-1}} \exp\left(s\Delta^i \left(1 - 2\eta^{i+1}\lambda_k\tau + (\eta^{i+1})^2 C^{i+1}\lambda_1 + s(\eta^{i+1})^2 8(b + \|\Sigma^*\|_F)^2\right)\right)$$

Since we choose the learning rate in Algorithm 1 such that

$$\eta^{i+1} < \frac{2\lambda_k\tau}{b(k+1)\lambda_1 + \frac{8}{1-\tau}\ln\frac{1}{\delta}(b + \|\Sigma^*\|_F)^2} = \frac{2\lambda_k\tau}{b(k+1)\lambda_1 + 8s(b + \|\Sigma^*\|_F)^2} \quad \text{(B.2)}$$

And since $\eta^{i+1} \le \frac{\sqrt{2}-1}{b}$, it can be seen that

$$C^{i+1} = kb + 2\eta^{i+1}b^2 + (\eta^{i+1})^2 b^3 \le b(k+1) \quad \text{(B.3)}$$

Combining Eq (B.2) and (B.3), we get

$$-2\eta^{i+1}\lambda_k\tau + (\eta^{i+1})^2 C^{i+1}\lambda_1 + s(\eta^{i+1})^2 8(b + \|\Sigma^*\|_F)^2 \leq 0$$

Therefore,

$$\mathbb{E}\left[\mathbb{1}_{\mathcal{G}_i}\exp\left(s\Delta^{i+1}\right)|\mathcal{F}_i\right] \leq \mathbb{1}_{\mathcal{G}_{i-1}}\exp\left(s\Delta^i\right)$$

Thus, letting $M_i = \mathbb{1}_{\mathcal{G}_{i-1}}\exp\left(s\Delta^i\right)$, $(M_i)_{i\geq 1}$ forms a supermartingale. A version of Doob's inequality for supermartingale (2, p. 231) implies that

$$\mathbb{P}\left(\mathcal{G}_t^c\right) = \mathbb{P}\left(\cup_{i=1}^t \mathcal{G}_{i-1}\setminus\mathcal{G}_i\right)$$

$$\leq \mathbb{P}\left(\sup_{i\geq 1}\mathbb{1}_{\mathcal{G}_{i-1}}\exp\left(s\Delta^i\right) > \exp\left(s(1-\tau)\right)\right) = \mathbb{P}\left(\sup_{i\geq 1} M_i > \exp\left(s(1-\tau)\right)\right)$$

$$\leq \frac{\mathbb{E}\left[M_1\right]}{\exp\left(s(1-\tau)\right)} = \frac{\mathbb{E}\left[\mathbb{1}_{\mathcal{G}_0}\exp\left(s\Delta^1\right)\right]}{\exp\left(s(1-\tau)\right)}$$

We bound the expectation as follows: By Inequality B.4 of Lemma 1,

$$\exp\left(s\Delta^1\right)\mathbb{1}_{\mathcal{G}_0} \leq \exp\left(s\left(\Delta^0(1-2\eta^1\lambda_k(1-\Delta^0)) - 2\eta^1 Z + (\eta^1)^2 C^1\lambda_1\Delta^0\right)\right)\mathbb{1}_{\mathcal{G}_0}$$

Taking expectation on both sides,

$$\mathbb{E}\left[\mathbb{1}_{\mathcal{G}_0}\exp\left(s\Delta^1\right)\right]$$

$$\leq \exp\left(s\left(\Delta^0(1-2\eta^1\lambda_k(1-\Delta^0)) + (\eta^1)^2 C^1\lambda_1\Delta^0\right)\right)\mathbb{E}\left[\exp\left(s(-2\eta^1 Z)\right)\right]$$

$$\leq \exp\left(s\left(\Delta^0(1-2\eta^1\lambda_k(1-\Delta^0)) + (\eta^1)^2 C^1\lambda_1\Delta^0\right)\right)\exp\left(2s^2(\eta^1)^2|Z|^2\right)$$

$$\leq \exp\left(s\Delta^0\left(1-2\eta^1\lambda_k\tau + (\eta^1)^2 C^1\lambda_1 + s(\eta^1)^2 8(b+\|\Sigma^*\|_F)^2\right)\right)$$

$$\leq \exp\left(s(1-\tau)\left(1-2\eta^1\lambda_k\tau + (\eta^1)^2 C^1\lambda_1 + s(\eta^1)^2 8(b+\|\Sigma^*\|_F)^2\right)\right)$$

$$\leq \exp\left(s(1-\tau)\right)$$

where the second inequality holds by Hoeffding's lemma (using the same argument as in Lemma 1), and the third and fourth inequality is by the fact that $\Delta^0 \leq (1-\tau)$ holds by our assumption. Finally,

$$\frac{\mathbb{E}\left[\mathbb{1}_{\mathcal{G}_0}\exp\left(s\Delta^1\right)\right]}{\exp\left(s(1-\tau)\right)} \leq \exp\left(-s(1-\tau)\right) \leq \delta\,,$$

since we set $s = \frac{1}{1-\tau}\ln\frac{1}{\delta}$. $\qquad\square$

## B.1 Auxiliary lemma for Proposition 1

*Proof of Lemma 1.* By V2 of Proposition 2, for $\Sigma^*$ with rank $k$,

$$tr(U^* P^{i+1}) \geq tr(U^* P^i)$$

$$+2\eta^{i+1}\sum_{\ell=1}^{k}\lambda_\ell(1-u_\ell^\top P^i u_\ell)(u_\ell^\top P^i u_\ell - \sum_{m\neq\ell}[1-u_m^\top P^i u_m]) + 2\eta^{i+1}Z$$
$$-(\eta^{i+1})^2 C^{i+1} tr(\Sigma^* - \Sigma^* P^i)$$

From this, we can derive

$$\Delta^{i+1} \leq \Delta^i - 2\eta^{i+1}\sum_{\ell=1}^{k}\lambda_\ell(1-u_\ell^\top P^t u_\ell)(1-\Delta^i) - 2\eta^{i+1}Z + (\eta^{i+1})^2 C^{i+1}\lambda_1\Delta^i$$
$$\leq \Delta^i - 2\eta^{i+1}\lambda_k tr(U^* - U^* P^i)(1-\Delta^i) - 2\eta^{i+1}Z + (\eta^{i+1})^2 C^{i+1}\lambda_1\Delta^i$$
$$= \Delta^i - 2\eta^{i+1}\lambda_k\Delta^i(1-\Delta^i) - 2\eta^{i+1}Z + (\eta^{i+1})^2 C^{i+1}\lambda_1\Delta^i$$
$$= \Delta^i(1-2\eta^{i+1}\lambda_k(1-\Delta^i)) - 2\eta^{i+1}Z + (\eta^{i+1})^2 C^{i+1}\lambda_1\Delta^i$$

This implies that for any $s > 0$,

$$\exp\left(s\Delta^{i+1}\right) \leq \exp\left(s\left(\Delta^i(1-2\eta^{i+1}\lambda_k(1-\Delta^i)) - 2\eta^{i+1}Z + (\eta^{i+1})^2 C^{i+1}\lambda_1\Delta^i\right)\right)$$

Multiplying both sides of the inequality by $\mathbb{1}_{\mathcal{G}_i}$, we get

$$\exp\left(s\Delta^{i+1}\right)\mathbb{1}_{\mathcal{G}_i}$$
$$\leq \exp\left(s\left(\Delta^i(1-2\eta^{i+1}\lambda_k(1-\Delta^i)) - 2\eta^{i+1}Z + (\eta^{i+1})^2 C^{i+1}\lambda_1\Delta^i\right)\right)\mathbb{1}_{\mathcal{G}_i} \quad \text{(B.4)}$$

We can further upper bound the RHS of Inequality (B.4) above as

$$\exp\left(s\left(\Delta^i(1-2\eta^{i+1}\lambda_k(1-\Delta^i)) - 2\eta^{i+1}Z + (\eta^{i+1})^2 C^{i+1}\lambda_1\Delta^i\right)\right)\mathbb{1}_{\mathcal{G}_i}$$
$$\leq \exp\left(s\left(\Delta^i(1-2\eta^{i+1}\lambda_k\tau) - 2\eta^{i+1}Z + (\eta^{i+1})^2 C^{i+1}\lambda_1\Delta^i\right)\right)\mathbb{1}_{\mathcal{G}_i}$$
$$\leq \exp\left(s\left(\Delta^i(1-2\eta^{i+1}\lambda_k\tau) - 2\eta^{i+1}Z + (\eta^{i+1})^2 C^{i+1}\lambda_1\Delta^i\right)\right)\mathbb{1}_{\mathcal{G}_{i-1}}$$
$$\leq \mathbb{1}_{\mathcal{G}_{i-1}}\exp\left(s\left(\Delta^i(1-2\eta^{i+1}\lambda_k\tau) + (\eta^{i+1})^2 C^{i+1}\lambda_1\Delta^i\right)\right)\exp\left(s\left(-2\eta^{i+1}Z\right)\right)$$

The first inequality is due to the fact that "$\{\mathbb{1}_{\mathcal{G}_i} = 1\}$ implies $\{\Delta^i \leq 1 - \tau\}$" and the second inequality holds since $\mathcal{G}_i \subset \mathcal{G}_{i-1}$. Incorporating this bound into inequality (B.4) and taking conditional expectation w.r.t. $\mathcal{F}_i$ on both sides, we get

$$\mathbb{1}_{\mathcal{G}_i}\mathbb{E}\left[\exp\left(s\Delta^{i+1}\right)|\mathcal{F}_i\right] = \mathbb{E}\left[\exp\left(s\Delta^{i+1}\right)\mathbb{1}_{\mathcal{G}_i}|\mathcal{F}_i\right]$$
$$\leq \mathbb{1}_{\mathcal{G}_{i-1}}\exp\left(s\left(\Delta^i(1-2\eta^{i+1}\lambda_k\tau) + (\eta^{i+1})^2 C^{i+1}\lambda_1\Delta^i\right)\right)\mathbb{E}\left[\exp\left(s\left(-2\eta^{i+1}Z\right)\right)|\mathcal{F}_i\right]$$

Now we upper bound $\mathbb{E}\left[\exp\left(s\left(-2\eta^{i+1}Z\right)\right)|\mathcal{F}_i\right]$ : Since

$$-2\eta^{i+1}|Z| \leq 2\eta^{i+1}(-Z) \leq 2\eta^{i+1}|Z|\,,$$

and
$$\mathbb{E}\left[2s\eta^{i+1}(-Z)|\mathcal{F}_i\right] = \mathbb{E}\left[2s\eta^{i+1}Z|\mathcal{F}_i\right] = 0\,,$$

by Hoeffding's lemma

$$\mathbb{E}\left[\exp\left(2s\eta^{i+1}(-Z)|\mathcal{F}_i\right)\right] \le \exp\left(\frac{s^2(4\eta^{i+1}|Z|)^2}{8}\right) = \exp\left(2s^2(\eta^{i+1})^2|Z|^2\right)\,.$$

Combining this with the previous bound, we get

$$\mathbb{1}_{\mathcal{G}_i}\,\mathbb{E}\left[\exp\left(s\Delta^{i+1}\right)|\mathcal{F}_i\right]$$
$$\le \mathbb{1}_{\mathcal{G}_{i-1}}\exp\left(s\left(\Delta^i(1-2\eta^{i+1}\lambda_k\tau)+(\eta^{i+1})^2C^{i+1}\lambda_1\Delta^i\right)\right)\exp\left(2s^2(\eta^{i+1})^2|Z|^2\right)$$

$\square$

# C   Proofs for Proposition 2

*Proof of Proposition 2.* We consider

$$\mathbb{E}\left[trU^*P^{t+1}\big|\mathcal{F}_t\right] = \mathbb{E}\left[trU^*(W^{t+1})^\top(W^{t+1}(W^{t+1})^\top)^{-1}W^{t+1}\big|\mathcal{F}_t\right]\,,$$

Since $U^*$ is positive semidefinite, we can write it as $U^* = ((U^*)^{1/2})^2$. By the proof of Lemma 2,

$$(W^{t+1}(W^{t+1})^\top)^{-1} \succeq (1-(\eta^{t+1})^2\|r^{t+1}\|^2\|s^{t+1}\|^2)I_{k'}$$

Letting $V := W^{t+1}(U^*)^{1/2}$, this implies that

$$V^\top\left[W^{t+1}(W^{t+1})^\top)^{-1} - (1-(\eta^{t+1})^2\|r^{t+1}\|^2\|s^{t+1}\|^2)I_{k'}\right]V \succeq 0$$

That is, the matrix on the left-hand-side above is positive semi-definite. Since trace of a positive semi-definite matrix is non-negative, we have

$$tr(V^\top W^{t+1}(W^{t+1})^\top)^{-1}V) \ge tr(V^\top(1-(\eta^{t+1})^2\|r^{t+1}\|^2\|s^{t+1}\|^2)V)$$

By commutative property of trace, we further get

$$tr(U^*(W^{t+1})^\top[W^{t+1}(W^{t+1})^\top]^{-1}W^{t+1}) = tr(V^\top W^{t+1}(W^{t+1})^\top)^{-1}V)$$
$$\ge tr(V^\top(1-(\eta^{t+1})^2\|r^{t+1}\|^2\|s^{t+1}\|^2)V)$$
$$= (1-(\eta^{t+1})^2\|r^{t+1}\|^2\|s^{t+1}\|^2)tr(U^*(W^{t+1})^\top W^{t+1})$$

Taking expectation on both sides, we get

$$\mathbb{E}\left[trU^*P^{t+1}\big|\mathcal{F}_t\right] \ge (1-(\eta^{t+1})^2\|r^{t+1}\|^2\|s^{t+1}\|^2)\,\mathbb{E}\left[tr(U^*(W^{t+1})^\top W^{t+1})\big|\mathcal{F}_t\right]$$

Now we in turn lower bound $\mathbb{E}\left[tr[U^*(W^{t+1})^\top W^{t+1}]\big|\mathcal{F}_t\right]$. First, we have

$$(W^{t+1})^\top W^{t+1} = (W^t + \eta^{t+1}s^{t+1}(r^{t+1})^\top)^\top(W^t + \eta^{t+1}s^{t+1}(r^{t+1})^\top)$$

$$= P^t + \eta^{t+1}r^{t+1}(s^{t+1})^\top W^t + \eta^{t+1}(W^t)^\top s^{t+1}(r^{t+1})^\top + (\eta^{t+1})^2\|s^{t+1}\|^2 r^{t+1}(r^{t+1})^\top$$

This implies that

$$
\begin{aligned}
\mathbb{E}\left[tr[U^*(W^{t+1})^\top W^{t+1}]\big|\mathcal{F}_t\right] &= tr(U^*\,\mathbb{E}\left[(W^{t+1})^\top W^{t+1}\big|\mathcal{F}_t\right]) \\
&= tr(U^*P^t) + \eta^{t+1}tr(\mathbb{E}\left[U^*r^{t+1}(s^{t+1})^\top\big|\mathcal{F}_t\right]W^t) \\
&\quad + \eta^{t+1}tr(\mathbb{E}\left[U^*(W^t)^\top s^{t+1}(r^{t+1})^\top\big|\mathcal{F}_t\right]) \\
&\quad + (\eta^{t+1})^2\,\mathbb{E}\left[\|s^{t+1}\|^2 tr(U^*r^{t+1}(r^{t+1})^\top)\big|\mathcal{F}_t\right] \\
&\geq tr(U^*P^t) + \eta^{t+1}tr(U^*\,\mathbb{E}\left[r^{t+1}(s^{t+1})^\top\big|\mathcal{F}_t\right]W^t) \\
&\quad + \eta^{t+1}tr(U^*\,\mathbb{E}\left[(W^t)^\top s^{t+1}(r^{t+1})^\top\big|\mathcal{F}_t\right]) \\
&\geq tr(U^*P^t) + 2\eta^{t+1}tr(U^*\,\mathbb{E}\left[(W^t)^\top s^{t+1}(r^{t+1})^\top\big|\mathcal{F}_t\right])
\end{aligned}
$$

the second to last inequality follows since we can drop the non-negative term, and the last inequality holds since the $tr(A) = tr(A^\top)$ for any square matrix $A$. Since

$$\mathbb{E}\left[s^{t+1}(r^{t+1})^\top\big|\mathcal{F}_t\right] = W^t(\Sigma^* - \Sigma^*P^t),$$

we have

$$trU^*\,\mathbb{E}\left[(W^t)^\top s^{t+1}(r^{t+1})^\top\big|\mathcal{F}_t\right] = trU^*(P^t\Sigma^* - P^t\Sigma^*P^t).$$

By Lemma 3,

$$
\begin{aligned}
trU^*\,\mathbb{E}\left[(W^t)^\top s^{t+1}(r^{t+1})^\top\big|\mathcal{F}_t\right] & \\
= trU^*(P^t\Sigma^* - P^t\Sigma^*P^t) & \\
\geq \sum_{i=1}^k \lambda_i(1 - u_i^\top P^t u_i)(u_i^\top P^t u_i - \sum_{j\neq i, j\in[k]}[1 - u_j^\top P^t u_j]) &
\end{aligned}
$$

Then we have,

$$\mathbb{E}\left[tr[U^*(W^{t+1})^\top W^{t+1}]\big|\mathcal{F}_t\right] \geq tr(U^*P^t)$$
$$+2\eta^{t+1}\sum_{i=1}^k \lambda_i(1 - u_i^\top P^t u_i)(u_i^\top P^t u_i - \sum_{j\neq i, j\in[k]}[1 - u_j^\top P^t u_j])$$

Now we can bound $\mathbb{E}\left[trU^*P^{t+1}\big|\mathcal{F}_t\right]$ as:

$$
\begin{aligned}
&\mathbb{E}\left[tr(U^*(W^{t+1})^\top[W^{t+1}(W^{t+1})^\top]^{-1}W^{t+1})\big|\mathcal{F}_t\right] \\
\geq{}& \mathbb{E}\left[tr(U^*(W^{t+1})^\top W^{t+1})\big|\mathcal{F}_t\right] - \mathbb{E}\left[(\eta^{t+1})^2\|r^{t+1}\|^2\|s^{t+1}\|^2 tr[U^*(W^{t+1})^\top W^{t+1}]\big|\mathcal{F}_t\right] \\
\geq{}& tr(U^*P^t) + 2\eta^{t+1}\sum_{i=1}^k \lambda_i(1 - u_i^\top P^t u_i)(u_i^\top P^t u_i - \sum_{j\neq i, j\in[k]}[1 - u_j^\top P^t u_j]) \\
&- \mathbb{E}\left[(\eta^{t+1})^2\|r^{t+1}\|^2\|s^{t+1}\|^2 tr(U^*(W^{t+1})^\top W^{t+1})\big|\mathcal{F}_t\right] \quad\text{(C.5)}
\end{aligned}
$$

Note that the second term in the inequality above can be lower bounded as:

$$\sum_{i=1}^{k} \lambda_i (1 - u_i^\top P^t u_i)(u_i^\top P^t u_i - \sum_{j \neq i, j \in [k]} [1 - u_j^\top P^t u_j])$$

$$= \sum_{i=1}^{k} \lambda_i (1 - u_i^\top P^t u_i)(\sum_{j \in [k]} u_j^\top P^t u_j - (k-1))$$

$$= \sum_{i=1}^{k} \lambda_i (1 - u_i^\top P^t u_i)(1 - \Delta^t) \geq \lambda_k \Delta^t (1 - \Delta^t)$$

Since $k' \leq d$, and rows of $W^t$ are orthonormal, we get

$$\|s^{t+1}\|^2 = \|W^t X^{t+1}\|^2 \leq \|X^{t+1}\|^2 .$$

Similarly, $\|r^{t+1}\|^2 \leq \|X^{t+1}\|^2$ . Therefore,

$$\|s^{t+1}\|^2 tr(U^*(W^{t+1})^\top W^{t+1})$$

$$\leq \|X^{t+1}\|^2 \left( trU^* P^t + 2\eta^{t+1} trU^* r^{t+1}(X^{t+1})^\top P^t + (\eta^{t+1})^2 \|s^{t+1}\|^2 trU^* r^{t+1}(r^{t+1})^\top \right)$$

$$= \|X^{t+1}\|^2 \left( trU^* P^t + 2\eta^{t+1}(X^{t+1})^\top P^t U^* r^{t+1} + (\eta^{t+1})^2 \|s^{t+1}\|^2 (r^{t+1})^\top U^* r^{t+1} \right)$$

$$\leq \|X^{t+1}\|^2 \left( trU^* P^t + 2\eta^{t+1} \|X^{t+1}\|^2 + (\eta^{t+1})^2 \|s^{t+1}\|^2 \|r^{t+1}\|^2 \right)$$

$$\leq \|X^{t+1}\|^2 \left( trU^* P^t + 2\eta^{t+1} \|X^{t+1}\|^2 + (\eta^{t+1})^2 \|X^{t+1}\|^4 \right)$$

On the other hand, we have

$$\mathbb{E}\left[\|r^{t+1}\|^2 \big| \mathcal{F}_t\right] = tr(\Sigma^* - \Sigma^* P^t)$$

Thus, the quadratic term (quadratic in $\eta^{t+1}$) in Eq (H.7) can be upper bounded as

$$\mathbb{E}\left[(\eta^{t+1})^2 \|r^{t+1}\|^2 \|s^{t+1}\|^2 tr(U^*(W^{t+1})^\top W^{t+1}) \big| \mathcal{F}_t\right]$$
$$\leq (\eta^{t+1})^2 C_o^t \, \mathbb{E}\left[\|r^{t+1}\|^2 \big| \mathcal{F}_t\right] = (\eta^{t+1})^2 C_o^t tr(\Sigma^* - \Sigma^* P^t)$$

where

$$C_o^t := \max_X \|X\|^2 \left( trU^* P^t + 2\eta^{t+1} \|X\|^2 + (\eta^{t+1} \|X\|^2)^2 \right)$$

$$\leq \max_X \|X\|^2 \left( k + 2\eta^{t+1} \|X\|^2 + (\eta^{t+1} \|X\|^2)^2 \right)$$

$$= kb + 2\eta^{t+1} b^2 + (\eta^{t+1})^2 b^3$$

Note that by our definition of $C^{t+1}$,

$$C^{t+1} := kb + 2\eta^{t+1} b^2 + (\eta^{t+1})^2 b^3$$

We get that

$$\mathbb{E}\left[(\eta^{t+1})^2\|r^{t+1}\|^2\|s^{t+1}\|^2 tr(U^*(W^{t+1})^\top W^{t+1})\big|\mathcal{F}_t\right] \leq C^{t+1}(\eta^{t+1})^2 tr(\Sigma^* - \Sigma^* P^t).$$

Since

$$\lambda_1 U^* \succeq \Sigma^* \succeq \lambda_k U^*$$

We have

$$(I - P^t)^\top \lambda_1 U^*(I - P^t) \succeq (I - P^t)^\top \Sigma^*(I - P^t)$$
$$\succeq (I - P^t)^\top \lambda_k U^*(I - P^t)$$

Note that the projection matrix satisfies $(I - P^t)^\top = (I - P^t)$ and $(I - P^t)(I - P^t) = (I - P^t)$. This implies that

$$\lambda_1 tr U^*(I - P^t) \geq tr\Sigma^*(I - P^t) \geq \lambda_k tr U^*(I - P^t) \tag{C.6}$$

Finally, plug the lower bound in Eq. (H.7) completes the proof:

$$\mathbb{E}\left[tr(U^* P^{t+1})\big|\mathcal{F}_t\right] \geq tr(U^* P^t) + \lambda_k \Delta^t(1 - \Delta^t) - (\eta^{t+1})^2 C^{t+1} \lambda_1 tr(U^*(I - P^t))$$
$$\geq tr(U^* P^t) + \lambda_k \Delta^t(1 - \Delta^t) - (\eta^{t+1})^2 C^{t+1} \lambda_1 \Delta^t$$

The inequality of the statement in Version 2 can be obtained similarly, by setting

$$Z = 2\left(tr(U^*(W^t)^\top s^{t+1}(r^{t+1})^\top) - \mathbb{E}\left[tr(U^*(W^t)^\top s^{t+1}(r^{t+1})^\top)\big|\mathcal{F}_t\right]\right)$$

It is clear that $\mathbb{E}\left[Z\big|\mathcal{F}_t\right] = 0$. Now we upper bound $|Z|$: Since

$$tr(U^*(W^t)^\top s^{t+1}(r^{t+1})^\top) = tr U^* P^t X^{t+1}(X^{t+1})^\top (I - P^t)$$

we get (subsequently, we denote $P^t$ by $P$, $X^{t+1}$ by $X$)

$$|Z| = |2tr(U^* P X X^\top (I - P)) - 2tr(U^* P \Sigma^*(I - P))|$$
$$= 2|tr(XX^\top - \Sigma^*)(I - P)U^* P| \leq 2\sqrt{\|XX^\top - \Sigma^*\|_F^2 \|(I - P)U^* P\|_F^2}$$

We first bound $\|(I - P)U^* P\|_F^2$,

$$\|(I - P)U^* P\|_F^2 \leq \|(I - P)U^*\|_F^2 = tr(U^* - U^* P)$$

where the first inequality is due to the fact that $P$ is a projection matrix so that norms are at best preserved if not smaller; the second inequality is also due to the fact that both $U^*$ and $I - P$ are projection matrices, and thus $(I - P)(I - P) = I - P$ and $U^* U^* = U^*$. Now we bound $\|XX^\top - \Sigma^*\|_F^2$:

$$\|XX^\top - \Sigma^*\|_F^2 = tr(XX^\top - \Sigma^*)^\top (XX^\top - \Sigma^*)$$

$$= \|X\|^4 - 2X^\top \Sigma^* X + \|\Sigma^*\|_F^2 \leq \|X\|^4 + \|\Sigma^*\|_F^2$$

where the last inequality is due to the fact that $\Sigma^*$ is positive semidefinite, that is, for any $x$, we have $x^\top \Sigma^* x \geq 0$. Finally,

$$|Z| \leq 2\sqrt{\|XX^\top - \Sigma^*\|_F^2 \|(I - P)U^* P\|_F^2}$$

$$\leq 2\sqrt{(\|X\|^4 + \|\Sigma^*\|_F^2) tr(U^* - U^* P)}$$

$$\leq 2(\|X\|^2 + \|\Sigma^*\|_F)\sqrt{\Delta^t} \leq 2(b + \|\Sigma^*\|_F)\sqrt{\Delta^t}$$

The third inequality is by the following argument: for any $a \geq 0, b \geq 0$, we have $\sqrt{a^2 + b^2} \leq a + b$, Letting $a = \|X\|^2$ and $b = \|\Sigma^*\|_F$ leads to the inequality. $\qquad \square$

## C.1 Auxiliary lemmas for Proposition 2

*Proof of Lemma 2.*

$$W^{t+1}(W^{t+1})^\top = (W^t + \eta^{t+1} s^{t+1}(r^{t+1})^\top)(W^t + \eta^{t+1} s^{t+1}(r^{t+1})^\top)^\top$$

$$= (W^t)(W^t)^\top + \eta^{t+1} s^{t+1}(r^{t+1})^\top (W^t)^\top$$

$$+ \eta^{t+1} W^t r^{t+1}(s^{t+1})^\top + (\eta^{t+1})^2 \|r^{t+1}\|^2 s^{t+1}(s^{t+1})^\top$$

$$= I_{k'} + (\eta^{t+1})^2 \|r^{t+1}\|^2 s^{t+1}(s^{t+1})^\top$$

where the last equality holds because $W^t$ has orthonormalized rows, and $r^{t+1}$ is orthogonal to rows of $W^t$. Let

$$E := (\eta^{t+1})^2 \|r^{t+1}\|^2 s^{t+1}(s^{t+1})^\top .$$

Note that $E$ is symmetric and positive semidefinite. We can eigen-decompose $E$ as

$$E = Q \Lambda Q^\top$$

where $Q$ is the eigenbasis and $\Lambda$ is a diagonal matrix with real non-negative diagonal values, with $\Lambda_{11} \geq \Lambda_{22}, \geq, \dots \Lambda_{k'k'}$, corresponding to the non-decreasing eigenvalues of $E$. Then

$$(I_{k'} + E)^{-1} = (QQ^\top + Q\Lambda Q^\top)^{-1} = Q(I_{k'} + \Lambda)^{-1} Q^\top ,$$

Since $I_{k'} + \Lambda$ is a diagonal matrix, for any $i \in [k']$, we have

$$(I_{k'} + \Lambda)_{ii}^{-1} = \frac{1}{1 + \Lambda_{ii}} \geq 1 - \Lambda_{ii} \geq 1 - \Lambda_{11}$$

This implies that the matrix

$$Q[(I_{k'} + \Lambda)^{-1} - (1 - \Lambda_{11})I_{k'}]Q^\top$$

is positive semidefinite, that is,

$$Q(I_{k'} + \Lambda)^{-1} Q^\top \succeq Q(1 - \Lambda_{11})I_{k'} Q^\top = (1 - \Lambda_{11})I_{k'}$$

Thus,

$$(W^{t+1}(W^{t+1})^\top)^{-1} = (I_{k'} + E)^{-1} \succeq (1 - \Lambda_{11})I_{k'}$$

Finally, we compute the largest eigenvalue of $E$, $\lambda_1(E) := \Lambda_{11}$:

$$\lambda_1(E) = \max_{\|y\|=1} y^\top E y = \max_{\|y\|=1} (\eta^{t+1})^2 \|r^{t+1}\|^2 (y^\top s^{t+1}(s^{t+1})^\top y)$$
$$= (\eta^{t+1})^2 \|r^{t+1}\|^2 \max_{\|y\|=1} (\langle s^{t+1}, y\rangle)^2 = (\eta^{t+1})^2 \|r^{t+1}\|^2 \|s^{t+1}\|^2$$

This completes the proof. $\qquad\qquad\qquad\qquad\qquad\qquad\qquad\qquad\qquad$ $\square$

*Proof of Lemma 3.* We first prove statement 1. Since $U^*$ is symmetric and positive semidefinite, we can write it as $U^* = ((U^*)^{1/2})^2$. So we have

$$tr(U^* - U^*P^t) = tr(U^*(I - P^t))$$
$$= tr((U^*)^{1/2}(I - P^t)(I - P^t)(U^*)^{1/2}) = \|(I - P^t)(U^*)^{1/2}\|_F^2$$

Therefore, $tr(U^*) = tr(U^*P^t)$ implies that

$$tr(U^* - U^*P^t) = \|(I - P^t)(U^*)^{1/2}\|_F^2 = 0$$

which implies

$$(I - P^t)(U^*)^{1/2} = 0$$

where "0" denotes the zero matrix. Thus,

$$\Gamma^t = tr(P^t\Sigma^*(I - P^t)U^*) = tr(P^t\Sigma^*(I - P^t)(U^*)^{1/2}(U^*)^{1/2}) = tr\,0 = 0\,.$$

Now we prove statement 2. First, we upper bound $tr(P^t\Sigma^*P^tU^*)$:

$$tr(P^t\Sigma^*P^tU^*) = tr(\sum_{p=1}^{k'}\sum_{i=1}^{k}\sum_{q=1}^{k'}\sum_{j=1}^{k}\lambda_i\langle w_p, u_i\rangle\langle w_q, u_i\rangle\langle w_q, u_j\rangle w_p u_j^\top)$$
$$= \sum_i \lambda_i \sum_j \sum_p \langle w_p, u_i\rangle\langle w_p, u_j\rangle \sum_q \langle w_q, u_i\rangle\langle w_q, u_j\rangle$$
$$= \sum_i \lambda_i \sum_j (u_i^\top P^t u_j)^2$$

Note that by Cauchy-Schwarz inequality,

$$(u_i^\top P^t u_j)^2 = (u_i^\top P^t(P^t)^\top u_j)^2 \le \|P^t u_i\|^2\|P^t u_j\|^2 = (u_i^\top P^t u_i)(u_j^\top P^t u_j)$$

On the other hand, for any $i$ and $j \neq i$ since $u_i \perp u_j$, we have

$$u_i^\top P^t u_j = u_i^\top u_j - u_i^\top(I - P^t)u_j = -u_i^\top(I - P^t)u_j$$

we have

$$(u_i^\top P^t u_j)^2 = (u_i^\top(I - P^t)u_j)^2 = (u_i^\top(I - P^t)(I - P^t)u_j)^2$$

$$\leq \|(I - P^t)u_i\|^2 \|(I - P^t)u_j\|^2$$
$$= (\|u_i\|^2 - \|P^t u_i\|^2)(\|u_j\|^2 - \|P^t u_j\|^2)$$
$$= (1 - u_i^\top P^t u_i)(1 - u_j^\top P^t u_j)$$

where the inequality is by Cauchy-Schwarz inequality, and the third equality is by combining orthogonality of projection $P^t$ and Pythagorean theorem. This implies that

$$tr(P^t \Sigma^* P^t U^*) = \sum_i \lambda_i \sum_j (u_i^\top P^t u_j)^2$$
$$= \sum_i \lambda_i (u_i^\top P^t u_i)^2 + \sum_i \lambda_i \sum_{j \neq i} (u_i^\top P^t u_j)^2$$
$$\leq \sum_i \lambda_i (u_i^\top P^t u_i)^2 + \sum_i \lambda_i \sum_{j \neq i} (1 - u_i^\top P^t u_i)(1 - u_j^\top P^t u_j)$$

Next, we expand $tr(P^t \Sigma^* U^*)$:

$$tr(P^t \Sigma^* U^*) = tr(U^* P^t \Sigma^*) = tr(\sum_i u_i u_i^\top P^t \sum_j \lambda_j u_j u_j^\top)$$
$$= \sum_i \sum_j \lambda_j u_i^\top P^t u_j u_i^\top u_j = \sum_i \lambda_i u_i^\top P^t u_i$$

Combining the upper bound on $tr(P^t \Sigma^* P^t U^*)$, we get,

$$tr(P^t \Sigma^* U^*) - tr(P^t \Sigma^* P^t U^*) = \sum_i \lambda_i u_i^\top P^t u_i - tr(P^t \Sigma^* P^t U^*)$$
$$\geq \sum_i \lambda_i u_i^\top P^t u_i - \sum_i \lambda_i (u_i^\top P^t u_i)^2 - \sum_i \lambda_i \sum_{j \neq i} (1 - u_i^\top P^t u_i)(1 - u_j^\top P^t u_j)$$
$$= \sum_i \lambda_i (1 - u_i^\top P^t u_i)(u_i^\top P^t u_i - \sum_{j \neq i} [1 - u_j^\top P^t u_j])$$

Recall that

$$\Delta^t = k - tr(U^* P^t) = k - \sum_{i=1}^k u_i^\top P^t u_i \,,$$

Therefore, the last term in the inequality above can be further lower bounded by $\lambda_k \Delta^t (1 - \Delta^t)$. $\qquad \square$

# D  Proof of Theorem 1

*Proof of Theorem 1.* Since by our assumption, $\Delta^o \leq 1 - \tau$, for any $\delta > 0$, and since we choose the learning rate such that

$$\eta \leq \min\{ \frac{2\lambda_k \tau}{\frac{8}{1-\tau} \ln \frac{1}{\delta} (b + \|\Sigma^*\|_F)^2 + b(k+1)\lambda_1}, \frac{\sqrt{2} - 1}{b} \} \,,$$

we can apply Proposition 1 to bound the probability of bad event, $\mathcal{G}_t^c$ as $\mathbb{P}\left(\mathcal{G}_t^c\right) \leq \delta$. By V1 of Proposition 2 (and let $C^{t+1}$ be as denoted therein),

$$\mathbb{E}\left[tr(U^*P^{t+1})|\mathcal{F}_t\right] \geq tr(U^*P^t) + 2\eta^{t+1}\lambda_k\Delta^t(1-\Delta^t) - (\eta^{t+1})^2 C^{t+1}\lambda_1\Delta^t \,,$$

Rearranging the inequality above and adding $k$ to both sides,

$$\mathbb{E}\left[\Delta^{t+1}|\mathcal{F}_t\right] \leq \Delta^t - 2\eta^{t+1}\lambda_k\Delta^t(1-\Delta^t) + (\eta^{t+1})^2 C^{t+1}\lambda_1\Delta^t$$
$$= \Delta^t\left(1 - 2\eta^{t+1}\lambda_k(1-\Delta^t) + (\eta^{t+1})^2 C^{t+1}\lambda_1\right),$$

Multiplying both sides of the inequality above by $\mathbb{1}_{\mathcal{G}_t}$, we get

$$\mathbb{E}\left[\Delta^{t+1}|\mathcal{F}_t\right]\mathbb{1}_{\mathcal{G}_t} \leq \Delta^t\left(1 - 2\eta^{t+1}\lambda_k(1-\Delta^t) + (\eta^{t+1})^2 C^{t+1}\lambda_1\right)\mathbb{1}_{\mathcal{G}_t}\,,$$

Since $\mathcal{G}_t$ is $\mathcal{F}_t$-measurable, we have

$$\mathbb{E}\left[\Delta^{t+1}\mathbb{1}_{\mathcal{G}_t}|\mathcal{F}_t\right] = \mathbb{E}\left[\Delta^{t+1}|\mathcal{F}_t\right]\mathbb{1}_{\mathcal{G}_t}\,,$$

When $\mathbb{1}_{\mathcal{G}_t} = 1$, we have $1 - \Delta^t \geq \tau$. Therefore,

$$\mathbb{E}\left[\Delta^{t+1}\mathbb{1}_{\mathcal{G}_t}|\mathcal{F}_t\right] \leq \Delta^t\left(1 - 2\eta^{t+1}\lambda_k\tau + (\eta^{t+1})^2 C^{t+1}\lambda_1\right)\mathbb{1}_{\mathcal{G}_t}$$
$$\leq \Delta^t\left(1 - 2\eta^{t+1}\lambda_k\tau + (\eta^{t+1})^2 C^{t+1}\lambda_1\right)\mathbb{1}_{\mathcal{G}_{t-1}}$$

where the last inequality holds since $\mathcal{G}_t \subset \mathcal{G}_{t-1}$. Taking expectation over both sides, we get the following recursion relation:

$$\mathbb{E}\left[\Delta^{t+1}\mathbb{1}_{\mathcal{G}_t}\right] \leq \mathbb{E}\left[\Delta^t\mathbb{1}_{\mathcal{G}_{t-1}}\right]\left(1 - 2\eta^{t+1}\lambda_k\tau + (\eta^{t+1})^2 C^{t+1}\lambda_1\right)$$

We further bound $1 - 2\eta^{t+1}\tau\lambda_k + (\eta^{t+1})^2 C^{t+1}\lambda_1$. First, note that since we require $\eta^{t+1} \leq \frac{\lambda_k\tau}{\lambda_1 b(k+3)}$, we get

$$\eta^{t+1}b \leq \frac{\lambda_k\tau}{\lambda_1(k+3)} \leq \frac{\tau}{(k+3)} \leq \frac{1}{k+3} \leq \frac{1}{4}.$$

and

$$C^{t+1} = b(k + 2\eta^{t+1}b + (\eta^{t+1})^2 b^2) \leq b(k+1)\,.$$

Thus, we get

$$1 - 2\eta^{t+1}\tau\lambda_k + (\eta^{t+1})^2 C^{t+1}\lambda_1 \leq 1 - 2\eta^{t+1}\tau\lambda_k + (\eta^{t+1})^2 b(k+1)\lambda_1$$

Since our requirement of $\eta^{t+1}$ also implies that

$$\eta^{t+1} \leq \frac{2\lambda_k\tau}{b(k+1)\lambda_1}\,,$$

it guarantees that

$$0 < 1 - 2\eta^{t+1}\tau\lambda_k + (\eta^{t+1})^2 b(k+1)\lambda_1 . < 1$$

For any $t$, define $\alpha^t := 2\eta^t\tau\lambda_k - (\eta^t)^2 b(k+1)\lambda_1$, we have

$$\mathbb{E}\left[\Delta^{t+1}\mathbb{1}_{\mathcal{G}_t}\right] \le \mathbb{E}\left[\Delta^t\mathbb{1}_{\mathcal{G}_{t-1}}\right]\left(1 - \alpha^{t+1}\right),$$

Recursively applying this relation, we get

$$\mathbb{E}\left[\Delta^{t+1}\mathbb{1}_{\mathcal{G}_t}\right] \le \Pi_{i=2}^{t+1}(1 - \alpha^i)\,\mathbb{E}\left[\Delta^1\mathbb{1}_{\mathcal{G}_0}\right]$$

Also note that

$$\Delta^1\mathbb{1}_{\mathcal{G}_0} \le (1 - \alpha^1)\Delta^0,$$

Therefore,

$$\mathbb{E}\left[\Delta^{t+1}\mathbb{1}_{\mathcal{G}_t}\right] \le \Pi_{i=1}^{t+1}(1 - \alpha^i)\Delta^0$$

Since for any $x \in (0,1)$, it holds that $\ln(1-x) \le -x$; we get

$$\Pi_{i=1}^t\left(1 - \alpha^i\right) \le \exp\left(-\sum_{i=1}^t \alpha^i\right)$$

Plugging in the value of $\alpha^i$'s, we get

$$\mathbb{E}\left[\Delta^t\mathbb{1}_{\mathcal{G}_{t-1}}\right] \le \exp\left(-\sum_{i=1}^t\left(2\eta^i\tau\lambda_k - (\eta^i)^2 b(k+1)\lambda_1\right)\right)$$

Again, by our requirement on learning rate, we have for any $t$

$$\eta^t \le \frac{\lambda_k\tau}{\lambda_1 b(k+3)} \le \frac{\lambda_k\tau}{\lambda_1 b(k+1)}$$

Thus,

$$2\eta^i\tau\lambda_k - (\eta^i)^2 b(k+1)\lambda_1 \ge \eta^i\tau\lambda_k > 0$$

Since we choose a constant learning rate $\eta$, this implies that

$$\mathbb{E}\left[\Delta^t\mathbb{1}_{\mathcal{G}_{t-1}}\right] \le \exp\left(-\sum_{i=1}^t \eta\tau\lambda_k\right) = \exp\left(-t\eta\tau\lambda_k\right)$$

Finally, since $\mathbb{1}_{\mathcal{G}_t} \le \mathbb{1}_{\mathcal{G}_{t-1}}$, we get

$$\mathbb{E}\left[\Delta^t\mathbb{1}_{\mathcal{G}_t}\right] \le \mathbb{E}\left[\Delta^t\mathbb{1}_{\mathcal{G}_{t-1}}\right] \le \exp\left(-t\eta\tau\lambda_k\right)$$

Combining this with the definition of conditional expectation, we get

$$\mathbb{E}\left[\Delta^t|\mathcal{G}_t\right] := \frac{\mathbb{E}\left[\Delta^t\mathbb{1}_{\mathcal{G}_t}\right]}{\mathbb{P}\left(\mathcal{G}_t\right)} \le \frac{\mathbb{E}\left[\Delta^t\mathbb{1}_{\mathcal{G}_t}\right]}{1-\delta} \le \frac{1}{1-\delta}\exp\left(-t\eta\tau\lambda_k\right)$$

where the first inequality is by our upper bound on the probability of bad event $\mathcal{G}_t^c$. $\qquad\square$

# E    Canonical (principal) angles between subspaces

**Definition 1 ((author?) (5)).** *Let $\mathcal{E}$ and $\mathcal{F}$ be $d$-dimensional subspaces of $\mathbb{R}^p$ with orthogonal projectors $E$ and $F$. Denote the singular values of $EF^{\perp}$ by $s_1 \geq s_2 \cdots \geq$. The canonical angles between $\mathcal{E}$ and $\mathcal{F}$ are the numbers*

$$\theta_k(\mathcal{E}, \mathcal{F}) = \arcsin(s_k)$$

*for $k = 1, \ldots, d$ and the angle operator between $\mathcal{E}$ and $\mathcal{F}$ is the $d \times d$ matrix*

$$\Theta(\mathcal{E}, \mathcal{F}) = diag(\theta_1, \ldots, \theta_d) \, .$$

*subject to*

$$\|x\| = \|y\| = 1, x^H x_i = 0, y^H y_i = 0, i = 1, \ldots, k - 1.$$

*The vectors $\{x_1, \ldots, x_m\}$ and $\{y_1, \ldots, y_m\}$ are called the principal vectors.*

**Proposition 3.** *Let $\mathcal{E}$ and $\mathcal{F}$ be $d$-dimensional subspaces of $\mathbb{R}^p$ with orthogonal projectors $E$ and $F$. Then The singular values of $EF^{\perp}$ are*

$$s_1, s_2, \ldots, s_d, 0, \ldots, 0.$$

*And*

$$\|\sin\Theta(\mathcal{E}, \mathcal{F})\|_F^2 = \|EF^{\perp}\|_F^2 \, .$$

# F    Random initialization guarantee

**Lemma 4.** *For any matrix $U^* = \sum_{i=1}^{k} u_i u_i^T$, where $u_i$'s have unit-norm and are orthogonal to each other. Suppose the entries of $W^o \in \mathbb{R}^{k' \times d}$ are drawn i.i.d. from $\mathcal{N}(0, 1)$. Let $P(W^o)$ be the orthogonal projection matrix into the subspace spanned by $W^o$. Then with probability at least $1 - 2k \exp\left(-(\varepsilon^2 - \varepsilon^3)k'/4\right) - \frac{k'+1}{dt^2}$,*

$$tr(U^* P(W^o)) \geq \frac{k'}{d(1+t)} k(1 - \varepsilon)$$

*Proof.* Let $\lambda_{\min}(A)$ and $\lambda_{\max}(A)$ denote the smallest and largest eigenvalue of a real symmetric matrix $A$, respectively. We have $P(W^o) = (W^o)^T (W^o (W^o)^T)^{-1} W^o$. Since the matrix $W^o (W^o)^T \in \mathbb{R}^{k' \times k'}$ is real, symmetric, and positive definite w.p. 1, there exists orthogonal matrix $Q \in \mathbb{R}^{k' \times k'}$ and diagonal matrix $\Lambda \in \mathbb{R}^{k' \times k'}$ such that

$$(W^o (W^o)^T = Q \Lambda Q^T$$

where $diag(\Lambda) = [\lambda_1, \ldots, \lambda_{k'}]$, the positive eigenvalues of $(W^o (W^o)^T$. Therefore,

$$\left(((W^o (W^o)^T)^{-1}\right) = Q \Lambda^{-1} Q^T$$

This implies that

$$\lambda_{\min}\left(((W^o (W^o)^T)^{-1}\right) = \frac{1}{\lambda_{\max}((W^o (W^o)^T))}$$

(It is possible that $W^o$ is rank deficient. But in this case, we can still define the pseudo-inverse of $(W^o(W^o)^T)$ to be of form

$$Q \begin{bmatrix} \Lambda_+^{-1} & 0 \\ 0 & 0 \end{bmatrix} Q^T$$

where $\Lambda_+$ has the positive eigenvalues of $\Lambda$. And we can proceed similarly by only considering $\lambda_{\min} of \Lambda_+$. ). For any $t > 0$,

$$\mathbb{P}\left(\lambda_{\min}\left(((W^o(W^o)^T)^{-1}\right) < \frac{1}{d(1+t)}\right) = \mathbb{P}\left(\lambda_{\max}(W^o(W^o)^T) > d + td\right)$$

$$= \mathbb{P}\left(\lambda_{\max}(\frac{1}{d}W^o(W^o)^T) > \frac{d+td}{d}\right)$$

Note that

$$\frac{1}{d}W^o(W^o)^T = \frac{1}{d}\sum_{i=1}^{d} x_i x_i^T \quad \text{where} \quad x_i \sim \mathcal{N}(0, I_{k'})$$

Thus, we can view $\frac{1}{d}W^o(W^o)^T$ as the sample covariance matrix of $\mathbb{E}\left[x_i x_i^T\right] = I_{k'}$. By Corollary 2.1 of (4), we get

$$\mathbb{P}\left(\frac{|\tilde{\lambda}_1 - \lambda_1|}{\lambda_1} \geq t\right) \leq \frac{1}{d}(\frac{k_1}{\lambda_1 t})^2 = \frac{1}{d}\frac{k_1^2}{\lambda_1^2 t^2}$$

where $\lambda_1 = \lambda_{\max}(I_{k'}) = 1$ and $\tilde{\lambda}_1 = \lambda_{\max}(\frac{1}{d}W^o(W^o)^T)$, $k_1 = (\mathbb{E}\left[\|x_i x_i^T y_1\|_2^2 - \lambda_1^2\right])^{1/2}$, and $y_1$ is the eigenvector corresponding to $\lambda_1$. Note that since $y_1$ can be any unit vector in $\mathbb{R}^{k'}$, for any random vector $x_i$, we can choose $y_1 = e_1$ (in fact, since Gaussian distribution is rotation-invariant, $x_i^T y_1$ has the same distribution for any $y_1$ on the unit-sphere), and

$$\mathbb{E}\left[\|x_i x_i^T y_1\|_2^2\right] = \mathbb{E}\left[(x_i^T e_1)^2 \|x_i\|^2\right] = \mathbb{E}\left[x_{i1}^2 \|x_i\|^2\right]$$

We define

$$Y(m) := \sum_{j=1}^{m} Z_j^2 \quad \text{for} \quad Z_j \sim \mathcal{N}(0, 1),$$

That is, $Y(m)$ is a random variable drawn from Chi-squared distribution with degree of freedom equal to $m$. Let $Y_1 \sim Y(1)$, $Y_2 \sim Y(k'-1)$ be drawn independently from two Chi-squared distributions. We have

$$\mathbb{E}\left[x_{i1}^2 \|x_i\|^2\right] = \mathbb{E}\left[Z_1^4 + Z_1^2 \sum_{j=1}^{k'-1} Z_j^2\right] = \mathbb{E}\left[Y_1^2\right] + \mathbb{E}\left[Y_1 Y_2\right]$$

$$= \mathbb{E}\left[(Y_1^2\right] + \mathbb{E}\left[Y_1\right]\mathbb{E}\left[Y_2\right] = 3 + k' - 1 = k' + 2$$

Then $k_1^2 = \mathbb{E}\left[\|x_i x_i^T y_1\|_2^2 - \lambda_1^2\right] \leq= k' + 2 - 1 = k' + 1$, and

$$\mathbb{P}\left(\frac{|\tilde{\lambda}_1 - \lambda_1|}{\lambda_1} \geq t\right) \leq \frac{1}{d}\frac{k'+1}{t^2}$$

Thus,

$$\mathbb{P}\left(\lambda_{\max}(\frac{1}{d}W^o(W^o)^T) > \frac{d+dt}{d}\right) \leq \mathbb{P}\left(\frac{|\tilde{\lambda}_1 - \lambda_1|}{\lambda_1} \geq t\right) \leq \frac{1}{d}\frac{k'+1}{t^2}$$

Therefore

$$\mathbb{P}\left(\lambda_{\min}\left(((W^o(W^o)^T)^{-1}\right) < \frac{1}{d(1+t)}\right) \leq \frac{k'+1}{dt^2}$$

Now, we lower bound $tr(U^*P(W^o))$:

$$tr(U^*P(W^o)) = \sum_1^k u_i^T(W^o)^T(W^o(W^o)^T)^{-1}W^o u_i$$

Thus, with probability no less than $1 - \frac{k'+1}{dt^2}$,

$$tr(U^*P(W^o)) \geq \frac{1}{d(t+1)}\sum_{i=1}^k u_i^T(W^o)^T W^o u_i = \frac{1}{d(t+1)}\sum_{i=1}^k \|W^o u_i\|_2^2$$

Now, by the norm-preserving property of the random Gaussian matrix $W^o$ (e.g., see Theorem 1.2 of (3)), w.p. at least $1 - 2k\exp\left(-(\varepsilon^2 - \varepsilon^3)k'/4\right)$, for all $u_i, i \in [k]$,

$$\|W^o u_i\|_2^2\| \geq k'(1-\varepsilon)$$

This implies that, w.p. at least $1 - 2k\exp\left(-(\varepsilon^2 - \varepsilon^3)k'/4\right) - \frac{k'+1}{dt^2}$,

$$tr(U^*P(W^o)) \geq \frac{1}{d(1+t)}k'k(1-\varepsilon)$$

$\square$

*Proof of Lemma 1.* For any $i$, $tr(U^*P(W_i^o)) = \sum_{\ell=1}^{k_i}(w_\ell)^T U^* w_\ell$, where $w_\ell$ are rows sampled from the last iterate of the previous epoch, $W_{i-1}^T$. Each row has equal probability to be drawn, so for all $\ell$,

$$\mathbb{E}\left[(w_\ell)^T U^* w_\ell\right] = \frac{1}{k_{i-1}}\sum_{m=1}^{k_{i-1}}(w_m)^T U^* w_m = \frac{1}{k_{i-1}}tr(U^*P(W_{i-1}^T))\,.$$

Hence,

$$\mathbb{E}\left[tr(U^*P(W_i^o))\right] = k_i\sum_{\ell=1}^{k_i}\mathbb{E}\left[(w_\ell)^T U^* w_\ell\right] = \frac{k_i}{k_{i-1}}tr(U^*P(W_{i-1}^T))\,,$$

where the expectation is taken over the randomness of uniform row-sampling operation. Now taking expectation again over the randomness of Matrix Krasulina completes the proof. $\square$

# G Full-rank extension: proofs of Theorem 2

In this section, we follow the same proof structure as that of Theorem 1 and adapts it to show linear convergence of Algorithm 1 in the general full-rank case.

## G.1 Extension of Prop. 1

**Proposition 4** (Extension of Prop 1). *Fix any $0 < \delta \leq \frac{1}{e}$. Let learning rate schedule in Algorithm 1 be $\eta^t = \frac{c}{t+t_o}$. Suppose $W^o$ is initialized such that $\Delta^o \leq \frac{1-\tau}{2}$ for some $\tau \in (0,1)$. If*

$$c \geq \frac{1}{(\lambda_k - \lambda_{k+1})\tau} , \quad and \ \ t_o \geq \max\{\frac{64Bc^2 \ln \frac{1}{\delta}}{(\Delta^o)^2}, 1\} .$$

*then $\mathbb{P}\left(\cup_{i\geq 1}\mathcal{G}_{i-1} \setminus \mathcal{G}_i\right) \leq \delta$.*

*Proof.* For simplicity of notation, for any $i$, we let $\mathbb{E}_i[e^{s\Delta^i}] := \mathbb{E}[e^{s\Delta^i} \mathbb{1}_{\mathcal{G}_{i-1}}]$. By Lemma 5, for any $s > 0$,

$$\mathbb{E}_i\{e^{s\Delta^i}\} \leq \mathbb{E}_i\{e^{s\{(1-\frac{\beta}{t_o+i})\Delta^{i-1}\}}\} \exp\left(\frac{sc^2B}{(t_o+i)^2} + \frac{s^2c^2B}{(t_o+i)^2}\right)$$

$$\leq \mathbb{E}_{i-1}\{e^{s^{(1)}\Delta^{i-1}}\} \exp\left(\frac{sc^2B}{(t_o+i)^2} + \frac{s^2c^2B}{(t_o+i)^2}\right)$$

where $s^{(1)} = s(1 - \frac{\beta}{t_o+i})$. Similarly, the following recurrence relation holds for $k = 0, \ldots, i$:

$$\mathbb{E}_{i-k}\{e^{s^{(k)}\Delta^{i-k}}\} \leq \mathbb{E}_{i-(k+1)}\{e^{s^{(k+1)}\Delta^{i-k-1}}\} \exp\left(\frac{s^{(k)}c^2B}{(t_o+i-k)^2} + \frac{(s^{(k)})^2c^2B}{(t_o+i-k)^2}\right)$$

where $s^{(0)} := s$, and for $k \geq 1$, $s^{(k)} := \Pi_{t=1}^k(1 - \frac{\beta}{t_o+(i-t+1)})s^{(0)}$. Note (see, e.g., (1)) that $\forall \beta > 0, k \geq 1$,

$$s^{(k)} = \Pi_{t=1}^k(1 - \frac{\beta}{t_o+(i-t+1)})s \leq (\frac{t_o+i-k+1}{t_o+i})^\beta s$$

Since the bound is shrinking as $\beta$ increases and $\beta = 2c(\lambda_k - \lambda_{k+1})\tau \geq 2$,

$$\frac{s^{(k)}}{(t_0+i-k)^2} \leq (\frac{t_o+i-k+1}{t_o+i})^2 \frac{s}{(t_o+i-k)^2} \leq \frac{4s}{(t_o+i)^2}$$

Repeatedly applying the relation, we get

$$\mathbb{E}_i\{e^{s\Delta^i}\} \leq e^{s^{(i)}\Delta^0} \exp\left(\sum_{k=0}^{i-1}(\frac{4sc^2B}{(t_o+i)^2} + \frac{4s^2c^2B}{(t_o+i)^2})\right)$$

$$\leq \exp\left(s(\frac{t_o+1}{t_o+i})^\beta \Delta^0 + (sc^2B + s^2c^2B)\frac{4i}{(t_o+i)^2}\right)$$

Then we can apply Markov's inequality, for any $s_i > 0$,

$$\mathbb{P}\left(\mathcal{G}_{i-1} \setminus \mathcal{G}_i\right) = \mathbb{P}\left(\Delta^i \mathbb{1}_{\mathcal{G}_{i-1}} > 1 - \tau\right) \leq \mathbb{P}\left(\Delta^i \mathbb{1}_{\mathcal{G}_{i-1}} > 2\Delta^o\right)$$

$$= \mathbb{P}\left(\mathbb{1}_{\mathcal{G}_{i-1}} e^{s_i \Delta^i} > e^{s_i 2\Delta^o}\right) \leq \frac{\mathbb{E}\left[e^{s_i \Delta^i} \mathbb{1}_{\mathcal{G}_{i-1}}\right]}{e^{s_i 2\Delta^o}} = \frac{\mathbb{E}_i[e^{s_i \Delta^i}]}{e^{s_i 2\Delta^o}}$$

Combining this with the upper bound on $\mathbb{E}_i \, e^{s_i \Delta^i}$, we get

$$\mathbb{P}\left(\mathcal{G}_{i-1} \setminus \mathcal{G}_i\right)$$

$$\leq \exp\left(-s_i\left(\Delta^o(2 - (\frac{t_o + 1}{t_o + i})^\beta) - (B + s_i B)\frac{4c^2 i}{(t_o + i)^2}\right)\right)$$

$$\leq \exp\left(-s_i\left(\Delta^o - (B + s_i B)\frac{4c^2 i}{(t_o + i)^2}\right)\right)$$

since $i \geq 1$. We choose $s_i = \frac{1}{\Delta} \ln \frac{(i+1)^2}{\delta}$ with $\Delta = \frac{\Delta^o}{2}$.

Recall that we choose

$$t_o \geq \max\{\frac{64Bc^2 \ln \frac{1}{\delta}}{(\Delta^o)^2}, 1\},$$

Since $\delta < \frac{1}{e}$, we have $t_o \geq \frac{64Bc^2}{(\Delta^o)^2} = \frac{16Bc^2}{\Delta^2}$. Note that

$$-\frac{\Delta(t_o + i)}{4Bc^2} + \frac{2\ln(i+1)^2/\delta}{\Delta} \leq -\frac{\Delta(t_o + i)}{4Bc^2} + \frac{\ln(i + t_o)^4/\delta}{\Delta}$$

Consider the function

$$f(y) = -\frac{\Delta y}{4Bc^2} + \frac{\ln y^4/\delta}{\Delta}$$

and its derivative

$$f'(y) = -\frac{\Delta}{4Bc^2} + \frac{4}{y\Delta}$$

$f'(y)$ is smaller than zero whenever $y \geq \frac{16Bc^2}{\Delta^2}$. And when $y = \frac{16Bc^2}{\Delta^2} \ln \frac{1}{\delta}$, $f(1) \leq 0$. Thus, for our choice of $t_o$,

$$1 + s_i \leq 2s_i \leq \frac{\Delta(t_o + i)}{4Bc^2},$$

which implies that

$$2\Delta - (B + s_i B)\frac{4c^2 i}{(t_o + i)^2} \geq 2\Delta - (B + s_i B)\frac{4c^2}{t_o + i} \geq \Delta$$

and hence,

$$\mathbb{P}\left(\mathcal{G}_{i-1} \setminus \mathcal{G}_i\right) \leq e^{-\frac{1}{\Delta}(\ln \frac{(1+i)^2}{\delta})\Delta} = \frac{\delta}{(i+1)^2}$$

Finally, we have

$$\mathbb{P}\left(\cup_{i \geq 1}\mathcal{G}_{i-1} \setminus \mathcal{G}_i\right) \leq \sum_{i=1}^{\infty} \mathbb{P}\left(\mathcal{G}_{i-1} \setminus \mathcal{G}_i\right) \leq \delta.$$

$\square$

## G.2 Auxiliary lemma for Prop. 4

**Lemma 5** (Supermartingale construction: full rank case). *Let $k$ and $b$ be defined as in Eq (2.4), and $\eta^t := \frac{c}{t+t_o}$ for $c, t_o > 0$. Suppose $\mathcal{G}_0$ holds.*

$$\text{Let } B := \max\left(8(b + \|\Sigma^*\|_F)^2 k, \left(kb + 2cb^2 + c^2b^3\right)\lambda_1(d-k)\right),$$

$$\beta := 2c(\lambda_k - \lambda_{k+1})\tau,$$

*Then for any $i \leq t$, and for any constant $s > 0$,*

$$\mathbb{E}\left[\mathbb{1}_{\mathcal{G}_i} \exp\left(s\Delta^{i+1}\right) | \mathcal{F}_i\right]$$

$$\leq \mathbb{1}_{\mathcal{G}_{i-1}} \exp\left(s\Delta^i\left(1 - \frac{\beta}{i+1+t_o}\right) + s(\frac{c}{i+1+t_o})^2 B + s^2(\frac{c}{i+1+t_o})^2 B\right).$$

*Proof.* By Prop 5

$$tr(U^* P^{i+1}) \geq tr(U^* P^i) + 2\eta^{i+1}(\lambda_k - \lambda_{k+1})\Delta^i(1 - \Delta^t) + 2\eta^{i+1}Z$$
$$- (\eta^{i+1})^2 C^{i+1} tr\Sigma^*(I - P)$$
$$\geq tr(U^* P^i) + 2\eta^{i+1}(\lambda_k - \lambda_{k+1})\Delta^i(1 - \Delta^t) + 2\eta^{i+1}Z$$
$$- (\eta^{i+1})^2 C^{i+1}\lambda_1(d - k)$$

Then we get

$$\Delta^{i+1} \leq \Delta^i - 2\eta^{i+1}(\lambda_k - \lambda_{k+1})\Delta^i\tau - 2\eta^{i+1}Z + (\eta^{i+1})^2 C^{i+1}\lambda_1(d-k)$$
$$= \Delta^i(1 - 2\eta^{i+1}(\lambda_k - \lambda_{k+1})\tau) - 2\eta^{i+1}Z + (\eta^{i+1})^2 C^{i+1}\lambda_1(d-k)$$

Following the same argument as in Lemma 1, we get

$$\mathbb{E}\left[\mathbb{1}_{\mathcal{G}_i} \exp\left(s\Delta^{i+1}\right) | \mathcal{F}_i\right]$$
$$\leq \mathbb{1}_{\mathcal{G}_{i-1}} \exp\left(s\Delta^i\left(1 - 2\eta^{i+1}(\lambda_k - \lambda_{k+1})\tau\right)\right)$$
$$\times \exp\left(s(\eta^{i+1})^2 C^{i+1}\lambda_1(d-k) + 2s^2(\eta^{i+1})^2|Z|^2\right).$$

Since by definition,

$$C^{i+1} := kb + 2\eta^{i+1}b^2 + (\eta^{i+1})^2 b^3 \leq kb + 2\eta^1 b^2 + (\eta^1)^2 b^3$$
$$\leq kb + 2\frac{c}{1+t_o}b^2 + (\frac{c}{1+t_o})^2 b^3$$

We get

$$C^{i+1}\lambda_1(d-k) \leq \left(kb + 2\frac{c}{1+t_o}b^2 + (\frac{c}{1+t_o})^2 b^3\right)\lambda_1(d-k) \leq B,$$

On the other hand, since

$$|Z| \leq 2\sqrt{\|XX^\top - \Sigma^*\|_F^2 \|(I-P)U^* P\|_F^2} \leq 2\sqrt{(\|X\|^4 + \|\Sigma^*\|_F^2)tr(U^* - U^* P)}$$

$$\leq 2(\|X\|^2 + \|\Sigma^*\|_F)\sqrt{tr(U^* - U^*P)} \leq 2(b + \|\Sigma^*\|_F)\sqrt{k}$$

We get

$$2|Z|^2 \leq 8(b + \|\Sigma^*\|_F)^2 k \leq B\,,$$

Plug this into the bound on expectation completes the proof. $\qquad\square$

# H   Full-rank extension of Prop. 2

This section extends Prop. 2 from rank-$k$ to the full rank case as stated below.

**Proposition 5** (Iteration-wise subspace improvement: full rank case)**.** *At the $t+1$-th iteration of Algorithm 1, the following holds:*

*(V1)  Let $C^t := kb + 2\eta^t b^2 + (\eta^t)^2 b^3$ . Then*

$$\mathbb{E}\left[tr(U^*P^{t+1})|\mathcal{F}_t\right] \geq tr(U^*P^t)$$
$$+2\eta^{t+1}(\lambda_k - \lambda_{k+1})\Delta^t(1 - \Delta^t) - (\eta^{t+1})^2 C^{t+1} tr(\Sigma^*(I - P))$$

*(V2)  There exists a random variable Z, with*

$$\mathbb{E}\left[Z|\mathcal{F}_t\right] = 0 \ \ and \ \ |Z| \leq 2(b + \|\Sigma^*\|_F)\sqrt{\Delta^t}$$

*such that*

$$tr(U^*P^{t+1}) \geq tr(U^*P^t) + 2\eta^{t+1}(\lambda_k - \lambda_{k+1})\Delta^t(1 - \Delta^t)$$
$$+2\eta^{t+1}Z - (\eta^{t+1})^2 C^{t+1} tr(\Sigma^*(I - P))$$

*Proof of Proposition 5.* The proof is similar to proof of Proposition 2, with modification of Lemma 6 and the variance term in Eq. C.6. We consider

$$\mathbb{E}\left[trU^*P^{t+1}\big|\mathcal{F}_t\right] = \mathbb{E}\left[trU^*(W^{t+1})^\top (W^{t+1}(W^{t+1})^\top)^{-1}W^{t+1}\big|\mathcal{F}_t\right]\,,$$

Since $U^*$ is positive semidefinite, we can write it as $U^* = ((U^*)^{1/2})^2$. By the proof of Lemma 2,

$$(W^{t+1}(W^{t+1})^\top)^{-1} \succeq (1 - (\eta^{t+1})^2\|r^{t+1}\|^2\|s^{t+1}\|^2)I_{k'}$$

Letting $V := W^{t+1}(U^*)^{1/2}$, this implies that

$$V^\top \left[W^{t+1}(W^{t+1})^\top)^{-1} - (1 - (\eta^{t+1})^2\|r^{t+1}\|^2\|s^{t+1}\|^2)I_{k'}\right]V \succeq 0$$

That is, the matrix on the left-hand-side above is positive semi-definite. Since trace of a positive semi-definite matrix is non-negative, we have

$$tr(V^\top W^{t+1}(W^{t+1})^\top)^{-1}V) \geq tr(V^\top(1 - (\eta^{t+1})^2\|r^{t+1}\|^2\|s^{t+1}\|^2)V)$$

By commutative property of trace, we further get

$$tr(U^*(W^{t+1})^\top[W^{t+1}(W^{t+1})^\top]^{-1}W^{t+1}) = tr(V^\top W^{t+1}(W^{t+1})^\top)^{-1}V)$$
$$\geq tr(V^\top(1 - (\eta^{t+1})^2\|r^{t+1}\|^2\|s^{t+1}\|^2)V)$$
$$= (1 - (\eta^{t+1})^2\|r^{t+1}\|^2\|s^{t+1}\|^2)tr(U^*(W^{t+1})^\top W^{t+1})$$

Taking expectation on both sides, we get

$$\mathbb{E}\left[trU^*P^{t+1}\big|\mathcal{F}_t\right] \geq (1 - (\eta^{t+1})^2\|r^{t+1}\|^2\|s^{t+1}\|^2)\,\mathbb{E}\left[tr(U^*(W^{t+1})^\top W^{t+1})\big|\mathcal{F}_t\right]$$

Now we in turn lower bound $\mathbb{E}\left[tr[U^*(W^{t+1})^\top W^{t+1}]\big|\mathcal{F}_t\right]$ . First, we have

$$(W^{t+1})^\top W^{t+1} = (W^t + \eta^{t+1}s^{t+1}(r^{t+1})^\top)^\top(W^t + \eta^{t+1}s^{t+1}(r^{t+1})^\top)$$
$$= P^t + \eta^{t+1}r^{t+1}(s^{t+1})^\top W^t + \eta^{t+1}(W^t)^\top s^{t+1}(r^{t+1})^\top + (\eta^{t+1})^2\|s^{t+1}\|^2 r^{t+1}(r^{t+1})^\top$$

This implies that

$$\mathbb{E}\left[tr[U^*(W^{t+1})^\top W^{t+1}]\big|\mathcal{F}_t\right] = tr(U^*\,\mathbb{E}\left[(W^{t+1})^\top W^{t+1}\big|\mathcal{F}_t\right])$$
$$= tr(U^*P^t) + \eta^{t+1}tr\,\mathbb{E}\left[U^*r^{t+1}(s^{t+1})^\top\big|\mathcal{F}_t\right]W^t$$
$$+\eta^{t+1}tr(\mathbb{E}\left[U^*(W^t)^\top s^{t+1}(r^{t+1})^\top\big|\mathcal{F}_t\right])$$
$$+(\eta^{t+1})^2\,\mathbb{E}\left[\|s^{t+1}\|^2 tr(U^*r^{t+1}(r^{t+1})^\top)\big|\mathcal{F}_t\right]$$
$$\geq tr(U^*P^t) + \eta^{t+1}trU^*\,\mathbb{E}\left[r^{t+1}(s^{t+1})^\top\big|\mathcal{F}_t\right]W^t$$
$$+\eta^{t+1}tr(U^*\,\mathbb{E}\left[(W^t)^\top s^{t+1}(r^{t+1})^\top\big|\mathcal{F}_t\right])$$
$$\geq tr(U^*P^t) + 2\eta^{t+1}tr(U^*\,\mathbb{E}\left[(W^t)^\top s^{t+1}(r^{t+1})^\top\big|\mathcal{F}_t\right])$$

the second to last inequality follows since we can drop the non-negative term, and the last inequality holds since the $tr(A) = tr(A^\top)$ for any square matrix $A$. Since

$$\mathbb{E}\left[s^{t+1}(r^{t+1})^\top\big|\mathcal{F}_t\right] = W^t(\Sigma^* - \Sigma^*P^t),$$

we have

$$trU^*\,\mathbb{E}\left[(W^t)^\top s^{t+1}(r^{t+1})^\top\big|\mathcal{F}_t\right] = trU^*(P^t\Sigma^* - P^t\Sigma^*P^t).$$

By Lemma 6,

$$trU^*\,\mathbb{E}\left[(W^t)^\top s^{t+1}(r^{t+1})^\top\big|\mathcal{F}_t\right] = trU^*(P^t\Sigma^* - P^t\Sigma^*P^t) \geq (\lambda_k - \lambda_{k+1})\Delta^t(1 - \Delta^t),$$

Then we have,

$$\mathbb{E}\left[tr[U^*(W^{t+1})^\top W^{t+1}]\big|\mathcal{F}_t\right] \geq tr(U^*P^t) + 2(\lambda_k - \lambda_{k+1})\Delta^t(1 - \Delta^t).$$

Now we can bound $\mathbb{E}\left[trU^*P^{t+1}\big|\mathcal{F}_t\right]$ as:

$$\mathbb{E}\left[tr(U^*(W^{t+1})^\top[W^{t+1}(W^{t+1})^\top]^{-1}W^{t+1})\big|\mathcal{F}_t\right]$$
$$\geq \mathbb{E}\left[tr(U^*(W^{t+1})^\top W^{t+1})\big|\mathcal{F}_t\right] - \mathbb{E}\left[(\eta^{t+1})^2\|r^{t+1}\|^2\|s^{t+1}\|^2 tr[U^*(W^{t+1})^\top W^{t+1}]\big|\mathcal{F}_t\right]$$

$$\geq tr(U^*P^t) + 2(\lambda_k - \lambda_{k+1})\Delta^t(1 - \Delta^t)$$
$$- \mathbb{E}\left[(\eta^{t+1})^2\|r^{t+1}\|^2\|s^{t+1}\|^2 tr(U^*(W^{t+1})^\top W^{t+1})\big|\mathcal{F}_t\right] \quad \text{(H.7)}$$

Since $k \leq d$, and rows of $W^t$ are orthonormal, we get

$$\|s^{t+1}\|^2 = \|W^t X^{t+1}\|^2 \leq \|X^{t+1}\|^2.$$

Similarly, $\|r^{t+1}\|^2 \leq \|X^{t+1}\|^2$. Therefore,

$$\|s^{t+1}\|^2 tr(U^*(W^{t+1})^\top W^{t+1})$$
$$\leq \|X^{t+1}\|^2 \left(trU^*P^t + 2\eta^{t+1}trU^*r^{t+1}(X^{t+1})^\top P^t + (\eta^{t+1})^2\|s^{t+1}\|^2 trU^*r^{t+1}(r^{t+1})^\top\right)$$
$$= \|X^{t+1}\|^2 \left(trU^*P^t + 2\eta^{t+1}(X^{t+1})^\top P^tU^*r^{t+1} + (\eta^{t+1})^2\|s^{t+1}\|^2(r^{t+1})^\top U^*r^{t+1}\right)$$
$$\leq \|X^{t+1}\|^2 \left(trU^*P^t + 2\eta^{t+1}\|X^{t+1}\|^2 + (\eta^{t+1})^2\|s^{t+1}\|^2\|r^{t+1}\|^2\right)$$
$$\leq \|X^{t+1}\|^2 \left(trU^*P^t + 2\eta^{t+1}\|X^{t+1}\|^2 + (\eta^{t+1})^2\|X^{t+1}\|^4\right)$$

On the other hand, we have

$$\mathbb{E}\left[\|r^{t+1}\|^2\big|\mathcal{F}_t\right] = tr(\Sigma^* - \Sigma^*P^t)$$

Thus, the quadratic term (quadratic in $\eta^{t+1}$) in Eq (H.7) can be upper bounded as

$$\mathbb{E}\left[(\eta^{t+1})^2\|r^{t+1}\|^2\|s^{t+1}\|^2 tr(U^*(W^{t+1})^\top W^{t+1})\big|\mathcal{F}_t\right]$$
$$\leq (\eta^{t+1})^2 C_o^t \mathbb{E}\left[\|r^{t+1}\|^2\big|\mathcal{F}_t\right] = (\eta^{t+1})^2 C_o^t tr(\Sigma^* - \Sigma^*P^t)$$

where

$$C_o^t := \max_X \|X\|^2\left(trU^*P^t + 2\eta^{t+1}\|X\|^2 + (\eta^{t+1}\|X\|^2)^2\right)$$
$$\leq \max_X \|X\|^2\left(k + 2\eta^{t+1}\|X\|^2 + (\eta^{t+1}\|X\|^2)^2\right)$$
$$= kb + 2\eta^{t+1}b^2 + (\eta^{t+1})^2b^3$$

Note that by our definition of $C^{t+1}$,

$$C^{t+1} := kb + 2\eta^{t+1}b^2 + (\eta^{t+1})^2b^3$$

We get that

$$\mathbb{E}\left[(\eta^{t+1})^2\|r^{t+1}\|^2\|s^{t+1}\|^2 tr(U^*(W^{t+1})^\top W^{t+1})\big|\mathcal{F}_t\right] \leq C^{t+1}(\eta^{t+1})^2 tr(\Sigma^* - \Sigma^*P^t).$$

Finally, plug this bound in Eq. (H.7) completes the proof:

$$\mathbb{E}\left[tr(U^*P^{t+1})|\mathcal{F}_t\right]$$
$$\geq tr(U^*P^t) + (\lambda_k - \lambda_{k+1})\Delta^t(1 - \Delta^t) - (\eta^{t+1})^2C^{t+1}tr(\Sigma^*(I - P^t))$$

The inequality of the statement in Version 2 can be obtained similarly, by setting

$$Z = 2\left(tr(U^*(W^t)^\top s^{t+1}(r^{t+1})^\top) - \mathbb{E}\left[tr(U^*(W^t)^\top s^{t+1}(r^{t+1})^\top)|\mathcal{F}_t\right]\right)$$

It is clear that $\mathbb{E}\left[Z|\mathcal{F}_t\right] = 0$. Now we upper bound $|Z|$: Since

$$tr(U^*(W^t)^\top s^{t+1}(r^{t+1})^\top) = trU^*P^t X^{t+1}(X^{t+1})^\top (I - P^t)$$

we get (subsequently, we denote $P^t$ by $P$, $X^{t+1}$ by $X$)

$$|Z| = |2tr(U^*PXX^\top(I - P)) - 2tr(U^*P\Sigma^*(I - P))|$$

$$= 2|tr(XX^\top - \Sigma^*)(I - P)U^*P| \le 2\sqrt{\|XX^\top - \Sigma^*\|_F^2\|(I - P)U^*P\|_F^2}$$

We first bound $\|(I - P)U^*P\|_F^2$,

$$\|(I - P)U^*P\|_F^2 \le \|(I - P)U^*\|_F^2 = tr(U^* - U^*P)$$

where the first inequality is due to the fact that $P$ is a projection matrix so that norms are at best preserved if not smaller; the second inequality is also due to the fact that both $U^*$ and $I - P$ are projection matrices, and thus $(I - P)(I - P) = I - P$ and $U^*U^* = U^*$. Now we bound $\|XX^\top - \Sigma^*\|_F^2$:

$$\|XX^\top - \Sigma^*\|_F^2 = tr(XX^\top - \Sigma^*)^\top(XX^\top - \Sigma^*)$$
$$= \|X\|^4 - 2X^\top\Sigma^*X + \|\Sigma^*\|_F^2 \le \|X\|^4 + \|\Sigma^*\|_F^2$$

where the last inequality is due to the fact that $\Sigma^*$ is positive semidefinite, that is, for any $x$, we have $x^\top\Sigma^*x \ge 0$. Finally,

$$|Z| \le 2\sqrt{\|XX^\top - \Sigma^*\|_F^2\|(I - P)U^*P\|_F^2}$$

$$\le 2\sqrt{(\|X\|^4 + \|\Sigma^*\|_F^2)tr(U^* - U^*P)}$$

$$\le 2(\|X\|^2 + \|\Sigma^*\|_F)\sqrt{\Delta^t} \le 2(b + \|\Sigma^*\|_F)\sqrt{\Delta^t}$$

The third inequality is by the following argument: For any $x \ge 0, y \ge 0$, we have $\sqrt{x^2 + y^2} \le x + y$, Letting $x = \|X\|^2$ and $y = \|\Sigma^*\|_F$ leads to the inequality. $\square$

## H.1 Auxiliary lemmas for Proposition 5

**Lemma 6** (extension of Lemma 3 to full-rank case)**.** *Let $\Sigma^*$ be of full rank, and $\Gamma^t := tr(U^*P^t\Sigma^*(I_d - P^t))$, Then the following holds:*

  1. *$tr(U^*P^t) = tr(U^*)$ implies that $\Gamma^t = 0$.*

2. $\Gamma^t \geq (\lambda_k - \lambda_{k+1})\Delta^t(1 - \Delta^t)$.

*Proof of Lemma 6.* We first prove statement 1. Since $U^*$ is symmetric and positive semidefinite, we can write it as $U^* = ((U^*)^{1/2})^2$. So we have

$$tr(U^* - U^*P^t) = tr(U^*(I - P^t))$$
$$= tr((U^*)^{1/2}(I - P^t)(I - P^t)(U^*)^{1/2}) = \|(I - P^t)(U^*)^{1/2}\|_F^2$$

Therefore, $tr(U^*) = tr(U^*P^t)$ implies that

$$tr(U^* - U^*P^t) = \|(I - P^t)(U^*)^{1/2}\|_F^2 = 0$$

which implies

$$(I - P^t)(U^*)^{1/2} = 0$$

where "0" denotes the zero matrix. Thus,

$$\Gamma^t = tr(P^t\Sigma^*(I - P^t)U^*) = tr(P^t\Sigma^*(I - P^t)(U^*)^{1/2}(U^*)^{1/2}) = tr0 = 0,$$

Now we prove statement 2. Let

$$\Sigma^*_{\leq k} := \sum_{i=1}^{k} \lambda_i u_i u_i^T \ \text{ and } \ \Sigma^*_{>k} := \sum_{j=k+1}^{d} \lambda_j u_j u_j^T,$$

We can decompose $\Sigma^*$ as

$$\Sigma^* = \Sigma^*_{\leq k} + \Sigma^*_{>k},$$

and we get

$$\Gamma^t = tr(U^*P^t\Sigma^*_{\leq k}(I_d - P^t)) + tr(U^*P^t\Sigma^*_{>k}(I_d - P^t))$$
$$= tr(U^*P^t\Sigma^*_{\leq k}(I_d - P^t)) - tr(U^*P^t\Sigma^*_{>k}P^t),$$

which is by the fact that

$$tr(U^*P^t\Sigma^*_{>k}) = tr(P^t\Sigma^*_{>k}U^*) = 0,$$

$$tr(U^*P^t\Sigma^*_{\leq k}(I_d - P^t)) = tr(U^*P^t \sum_{i \leq k} \lambda_i u_i u_i^T(I_d - P^t))$$

$$= \sum_{i \leq k} \lambda_i tr(U^*P^t u_i u_i^T(I_d - P^t)) = \sum_{i \leq k} \lambda_i [tr(u_i^T P^t u_i) - tr(U^*P^t u_i u_i^T P^t)]$$

$$= \sum_{i \leq k} \lambda_i tr(u_i^T P^t(I - U^*)P^t u_i) = \sum_{i \leq k} \lambda_i tr(((I - U^*)P^t u_i)^T(I - U^*)P^t u_i)$$

Therefore, each term $tr(u_i^T P^t(I - U^*)P^t u_i)$ is non-negative, and we have

$$\lambda_i tr(u_i^T P^t(I - U^*)P^t u_i) \geq \lambda_k tr(u_i^T P^t(I - U^*)P^t u_i)$$

Implying that

$$tr(U^* P^t \Sigma^*_{\leq k}(I_d - P^t)) \geq \lambda_k \sum_{i \leq k} tr(u_i^T P^t (I - U^*) P^t u_i)$$
$$= \lambda_k tr(P^t (I - U^*) P^t U^*)$$

On the other hand,

$$tr(U^* P^t \Sigma^*_{>k} P^t) = tr(U^* P^t \sum_{j>k} \lambda_j u_j u_j^T P^t)$$

$$= \sum_{j>k} \lambda_j tr(U^* P^t u_j u_j^T P^t) = \sum_{j>k} \lambda_j tr(u_j^T P^t U^* P^t u_j)$$

$$= \sum_{j>k} \lambda_j tr((U^* P^t u_j)^T U^* P^t u_j) \leq \sum_{j>k} \lambda_{k+1} tr((U^* P^t u_j)^T U^* P^t u_j)$$

$$= \lambda_{k+1} \sum_{j>k} tr(u_j^T P^t U^* P^t u_j) = \lambda_{k+1} tr(P^t U^*_{>k} P^t U^*) = \lambda_{k+1} tr(P^t (I - U^*) P^t U^*)$$

where the upper bound holds because the terms $tr((U^* P^t u_j)^T U^* P^t u_j)$ are non-negative, and the last equality is by Lemma 7. Thus,

$$\Gamma^t \geq (\lambda_k - \lambda_{k+1}) tr(P^t (I - U^*) P^t U^*)$$

Finally, we apply the result in Lemma 3 to lower bound $tr(P^t(I - U^*)P^t U^*)$: In Lemma 3, it was shown that

$$tr(P \Sigma^*_{\leq k}(I - P) U^*)$$
$$\geq \sum_{i \leq k} \lambda_i (1 - u_i^T P^t u_i)(u_i^T P^t u_i - \Sigma_{j \neq i}[1 - u_j^T P^t u_j])$$

If we set all eigenvalues $\lambda_1, \ldots, \lambda_k$ in $\Sigma_{\leq k}$ to be 1, then $\Sigma^*_{\leq k}$ becomes $U^*$. So by the inequality above,

$$tr(P^t U^*(I - P^t) U^*)$$
$$\geq \sum_{i \leq k} (1 - u_i^T P^t u_i)(u_i^T P^t u_i - \Sigma_{j \neq i}[1 - u_j^T P^t u_j])$$
$$= \Delta^t (1 - \Delta^t)$$

Also since

$$tr(P^t U^*(I - P^t) U^*) = tr(P^t U^* U^*) - tr(P^t U^* P^t U^*) = tr(P^t U^*) - tr(P^t U^* P^t U^*)$$
$$= tr(P^t U^* P^t) - tr(P^t U^* P^t U^*) = tr(P^t U^* P^t (I - U^*)),$$

So we get

$$\Gamma^t \geq (\lambda_k - \lambda_{k+1}) tr(P^t (I - U^*) P^t U^*) \geq (\lambda_k - \lambda_{k+1}) \Delta^t (1 - \Delta^t)$$

$\square$

**Lemma 7** (Matrix equality). *Let $U^*_{>k} := \sum_{i=k+1}^{d} u_i u_i^T$, and $U^* := \sum_{i<k}^{d} u_j u_j^T$ such that $\{u_1, \ldots, u_d\}$ forms a orthonormal basis of $\mathbb{R}^d$. Then $U^*_{>k} = I - U^*$.*

*Proof.* To prove that $U^*_{>k} = I - U^*$, we only need to find an invertible matrix $R$ and show that

$$U^*_{>k} R = (I - U^*) R$$

Then it implies that

$$U^*_{>k} = U^*_{>k} R R^{-1} = (I - U^*) R R^{-1} = (I - U^*)$$

Note that $\{u_1, \ldots, u_d\}$ forms a orthonormal basis, so letting

$$R := \begin{pmatrix} \uparrow & \cdots & \uparrow \\ u_1 & \cdots & u_d \\ \downarrow & \cdots & \downarrow \end{pmatrix} \quad \text{and} \quad x = \begin{pmatrix} u_1^T y \\ \cdots \\ u_d^T y \end{pmatrix},$$

we have $RR^T = R^T R = I$. Let $M_{\star i}$ denote the $i$-th column of matrix $M$, then

$$(U^*_{>k} R)_{\star i} = \sum_{j>k} u_j u_j^T u_i \begin{cases} 0, & \text{for } i \le j \\ u_i, & \text{for } i > j \end{cases}$$

And

$$((I - U^*) R)_{\star i} = u_i - \sum_{j \le k} u_j u_j^T u_i = \begin{cases} 0, & \text{for } i \le j \\ u_i, & \text{for } i > j \end{cases}$$

Thus, $U^*_{>k} R = (I - U^*) R$, which finishes the proof. $\qquad\square$

## H.2 Proof of Theorem 2

*Proof.* By our choice of learning rate and the initialization condition, the conditions in Proposition 4 holds for any $t > i$, we apply it and get

$$\mathbb{P}\left(\mathcal{G}_t\right) \ge 1 - \delta,$$

This proves the first statement. In contrast to the proof of Theorem 1, by Prop. 5, the recurrence relation in the general case (that is, without low-rank assumption) is

$$tr U^* P^t \ge \mathbb{1}_{\mathcal{G}_{t-1}} (tr(U^* P^{t-1})$$
$$+ 2\eta^t (\lambda_k - \lambda_{k+1}) \Delta^{t-1} (1 - \Delta^{t-1}) + 2\eta^t Z - (\eta^t)^2 C^t tr \Sigma^* (I - P))$$

This implies that

$$\Delta^t \le \Delta^{t-1} - 2\eta^t (\lambda_k - \lambda_{k+1}) \Delta^{t-1} (1 - \Delta^{t-1}) + 2\eta^t Z + (\eta^t)^2 C^t \lambda_1 (d - k)]$$
$$\le \Delta^{t-1} - 2\eta^t (\lambda_k - \lambda_{k+1}) \Delta^{t-1} (1 - \Delta^{t-1}) + 2\eta^t Z + (\eta^t)^2 B]$$

We can multiply both sides by indicator variable $\mathbb{1}_{\mathcal{G}_{t-1}}$ and get

$$\mathbb{1}_{\mathcal{G}_{t-1}} \Delta^t \le \mathbb{1}_{\mathcal{G}_{t-1}} [\Delta^{t-1} - 2\eta^t (\lambda_k - \lambda_{k+1}) \Delta^{t-1} (1 - \Delta^{t-1}) + 2\eta^t Z + (\eta^t)^2 B]$$

$$\leq \mathbb{1}_{\mathcal{G}_{t-1}} \Delta^{t-1} - 2\eta^t(\lambda_k - \lambda_{k+1}) \, \mathbb{1}_{\mathcal{G}_{t-1}} \Delta^{t-1}\tau + 2\eta^t Z \, \mathbb{1}_{\mathcal{G}_{t-1}} + (\eta^t)^2 B \, \mathbb{1}_{\mathcal{G}_{t-1}}$$
$$\leq \mathbb{1}_{\mathcal{G}_{t-1}} \Delta^{t-1} - 2\eta^t(\lambda_k - \lambda_{k+1}) \, \mathbb{1}_{\mathcal{G}_{t-1}} \Delta^{t-1}\tau + 2\eta^t Z + (\eta^t)^2 B$$

where the second inequality holds since $\mathcal{G}_{t-1}$ holds implies that $\Delta^t < 1 - \tau$. Let $\beta$ be defined as in Lemma 5, this implies that

$$\mathbb{1}_{\mathcal{G}_{t-1}} \Delta^t \leq \mathbb{1}_{\mathcal{G}_{t-1}} \Delta^{t-1}(1 - \frac{\beta}{t_o + t}) + (\frac{c}{t_o + t})^2 B + 2\eta^t Z$$

Taking expectation, we get

$$\mathbb{E}[\mathbb{1}_{\mathcal{G}_{t-1}} \Delta^t] \leq \mathbb{E}[\mathbb{1}_{\mathcal{G}_{t-1}} \Delta^{t-1}](1 - \frac{\beta}{t_o + t}) + \frac{c^2 B}{(t + t_o)^2}$$

$$\leq \mathbb{E}[\mathbb{1}_{\mathcal{G}_{t-2}} \Delta^{t-1}](1 - \frac{\beta}{t_o + t}) + \frac{c^2 B}{(t + t_o)^2}$$

We subsequently denote by $\mathbb{E}_t[\cdot] := \mathbb{E}[\mathbb{1}_{\mathcal{G}_{t-1}} \cdot]$ for convenience. So the above inequality can be re-written as

$$\mathbb{E}_t[\Delta^t] \leq \mathbb{E}_{t-1}[\Delta^{t-1}](1 - \frac{\beta}{t_o + t}) + \frac{c^2 B}{(t + t_o)^2}$$

Since by our choice of parameter $c$ in the learning rate, $\beta = 2(\lambda_k - \lambda_{k+1})\tau c > 1$, we can apply Lemma 8 by letting $u_t \leftarrow \mathbb{E}_{t+t_o}[\Delta^{t+t_o}]$ (we temporarily change the notation $\mathbb{E}_t[\Delta^t]$ to $\mathbb{E}_{t+t_o}[\Delta^{t+t_o}]$ to match the notation in Lemma 8), $t \leftarrow t_o + t$, $a \leftarrow \beta$, and $b \leftarrow c^2 B$

$$\mathbb{E}_t[\Delta^t] \leq (\frac{t_o + 1}{t_o + t + 1})^\beta \Delta^o + \frac{c^2 B}{\beta - 1}(\frac{t_o + 2}{t_o + 1})^{\beta+1}\frac{1}{t_o + t + 1}$$

Also, since $(\lambda_k - \lambda_{k+1})\tau c > 1$, $\beta - 1 > (\lambda_k - \lambda_{k+1})\tau c$, and $\beta > 2$. Thus

$$\mathbb{E}_t[\Delta^t] \leq (\frac{t_o + 1}{t_o + t + 1})^2 \Delta^o + \frac{c^2 B}{(\lambda_k - \lambda_{k+1})\tau c}(\frac{t_o + 2}{t_o + 1})^{\beta+1}\frac{1}{t_o + t + 1}$$

$$= (\frac{t_o + 1}{t_o + t + 1})^2 \Delta^o + \frac{cB}{(\lambda_k - \lambda_{k+1})\tau}(\frac{t_o + 2}{t_o + 1})^{\beta+1}\frac{1}{t_o + t + 1}$$

Finally, note that

$$\mathbb{E}\left[\Delta^t | \mathcal{G}_t\right] = \frac{\mathbb{E}_t[\Delta^t]}{\mathbb{P}(\mathcal{G}_t)}$$

$$\leq \frac{1}{1 - \delta}\{(\frac{t_o + 1}{t_o + t + 1})^2 \Delta^o + \frac{cB}{(\lambda_k - \lambda_{k+1})\tau}(\frac{t_o + 2}{t_o + 1})^{\beta+1}\frac{1}{t_o + t + 1}\}.$$

$\square$

**Lemma 8** (Lemma D1 of (1)). *Consider a nonnegative sequence $(u_t : t \geq t_o)$, such that for some constants $a, b > 0$ and for all $t > t_o \geq 0$, $u_t \leq (1 - \frac{a}{t})u_{t-1} + \frac{b}{t^2}$. Then, if $a > 1$,*

$$u_t \leq (\frac{t_o + 1}{t + 1})^a u_{t_o} + \frac{b}{a - 1}(1 + \frac{1}{t_o + 1})^{a+1}\frac{1}{t + 1}$$