[Reviews · NeurIPS 2019]

Reviewer 1



POST-REBUTTAL UPDATE: I am happy with the authors' response to my main question. I maintain that this is a top paper and it should absolutely be accepted. ----------------- This was the best of the 6 papers I reviewed. Intuitive results, great idea and execution on the main result. The writing is very clear the discussion on related work thorough and detailed, and the paper overall is a joy to read. I believe that it should be accepted. Some critiques I had about the limitations of the proposed method (warm start which needs to use an initialization of rank proportional to the ambient dimension) are already discussed honestly in the paper, which also includes an effort to propose a heuristic method to “bootstrap” from a high-rank initialization progressively reducing the rank in a logarithmic number of phases. My main remaining question is the following: Your main result is given with high probability, in expectation. Could you elaborate with some discussion on what that guarantee means, and what it does not in terms of the actual convergence of the algorithm? In my opinion that’s the only piece of discussion missing from an otherwise great paper. Originality: The paper is motivated by a nuance in the known information-theoretic lower bounds for online stochastic PCA. The bounds suggest that the best possible rate is O(1/n), where n is the number of examples, but it critically relies on an assumption on the rank of the data. The authors point out how, for low rank matrices, this lower bound becomes uninformative, leaving the rank-constrain problem wide-open for improvement. The authors go on to do exactly that by proposing an algorithm that achieves exponentially fast convergence. Even though I have done some work in the area, I admit that I was unaware of this nuance in the lower bound. I have not seen other work exploit it in a similar way. Quality: The quality of the paper is excellent. From organization, writing, literature review, motivation and presentation of the results it is very well done and polished. Clarity: Excellent. I will recommend this paper to people interested in getting in the area and wanting to learn. Significance: Given the originality of the contributions 1 and 2 above, I think this paper can have a significant impact on the research community interested in similar problems. Minor typo, Line 252: “shaper” -> “sharper”

Reviewer 2



Originality: The method is novel and non-obvious. The original Krasulina method is literally SGD on the squared reconstruction error. However the generalization presented appears to be distinct from this form of generalization, and is instead motivated to preserve self-regulating variance decay as the residual decays. The relationship to related work, particularly VR-PCA, is explored in depth and adequately cited. Quality: The body of the submission is technically sound, and I have no reason to doubt the validity of the proofs in the supplement; the structure is reasonable, though I have not checked the arguments in detail. The authors are careful and honest about addressing the strengths and weaknesses of the method, in particular proposing Algorithm 2 to overcome a weakness in the assumptions in Theorem 1, acknowledging the desirability of extending the theoretical results to effectively low-rank data, and providing a detailed comparison with VR-PCA in theory and practice. However, the significance of this comparison would be strengthened by considering runtime, not just epochs. Clarity: The writing is very clear and well-organized. The introductory overview of approaches to PCA provides the reader with helpful orientation. There are two natural questions for which I’d appreciate further clarification: Significance: The results are highly significant given the centrality of PCA in data science and the modern context of online algorithms on big or streaming data. The method improves upon existing theory and practice.

Reviewer 3



***AFTER REBUTTAL*** After reading the response from the authors and the other reviewers' comments, I increase my score from 7 to 8. ***END OF COMMENT*** SUMMARY: The paper proposes an online stochastic learning algorithm to solve the non-convex problem of PCA. The proposed approach employs Krasulina’s method, which is generalised from vector to matrix representation. The idea is to demonstrate that the generalised Krasulina’s method is a stochastic gradient descent with a self-regulated gradient that can be tailored to solve the PCA problem. In addition to presenting a new application for Krasulina’s method, convergence rates are derived for low-rank and full-rank data. The derivation of the exponential convergence rate for low-rank data, in theorem 1, highlights that the convergence rate is actually affected by the intrinsic dimension, or number of components, and not by the dimensionality of the data. ORIGINALITY: The proposed method combines and tailors an old technique for a new purpose. It advances the research on online learning solutions to the PCA problem in the sense that the state-of-the-art approach, that combines the classic Oja’s method with a variance reduction step, is not an online algorithm because it requires a full pass over the dataset. The previous work in this field is well-referenced and accurately presented. QUALITY: The theoretical presentation is complete and justified with proofs. The empirical evaluation is performed in terms of a simulation study and two real-world experiments. In the real-world data experiments, the proposed algorithm is evaluated against the variance reduction augmented Oja’s method. In the simulation study, theorem 1 is shown to hold in practice. In the real-world experiments, the proposed approach is shown to converge faster before the competing method reaches a full pass. CLARITY: The paper is clearly written and well-organised. The results can probably be reproduced with the help of the pseudocode. SIGNIFICANCE: For practitioners working with large datasets, the presented method can be useful to reduce the dimensionality of the data.

[Author Response · NeurIPS 2019]

# 1 Rebuttal

We thank all reviewers for your positive and insightful feedback. Below are our answers to your questions.

## 3 R1

"Your main result is given with high probability, in expectation. Could you elaborate with some discussion on what that
guarantee means, and what it does not in terms of the actual convergence of the algorithm? In my opinion that's the
only piece of discussion missing from an otherwise great paper."

Thanks for pointing this out: In short, $\mathcal{G}_t$ in Theorem 1 can be interpreted as a "success event" (the event that the
algorithm's iterates up to time $t$ all stay within the "basin of attraction"). Our analysis shows that $\mathcal{G}_t$ holds with high
probability, and the expectation over $\Delta^t$ is restricted to this success event.

The same type of guarantee was used in [9] (Balsubramani et al).

Due to space limit, we have moved the definition of $\mathcal{G}_t$ to the Appendix before submission. We will add an explanation
on this topic to our paper.

## 13 R2

"However, the significance of this comparison would be strengthened by considering runtime, not just epochs."

This is indeed a good point and will be very helpful for practitioners who would like to try out this method. We will add
experiment that compares runtime.

## 17 R3

"(1) Why was the VR-PCA method not included in the simulation study?"

Our goal for the simulation study was to validate our theoretical result about Matrix Krasulina (exponential convergence
rate on low-rank data), to see how its convergence rate is affected by both the intrinsic and the data dimension, and
also to check whether the algorithm attains fast convergence (if not exactly exponential) when the data is effectively
low-rank, to empirically test out our conjecture about the algorithm.

"(2) It would have been interesting to see some computation times."

Thanks for pointing this out. We'll add experiments to compare runtime as well.

[Meta-Review · NeurIPS 2019]

This paper presents strong novel results on convergence of an online PCA algorithm. It is typically difficult to make such strong progress in an area as well-studied as online PCA, which is what makes this paper an impressive contribution to NeurIPS. The fundamental contributions are: 1) A variance analysis of Krasulina’s method, which at the time of Oja’s method’s development was the main contender. In recent years, Oja’s has been studied much more extensively, but it was never clear which is really the right approach. This paper gives strong argument that Krasulina’s method may be the stronger one when data are low rank. The paper states that it’s an open question whether this is also true for effectively low rank data, but the reviewers and I believe this is an important first step toward proving such a convergence result. 2) The authors point out that for exactly low-rank data, the standard information-theoretic lower bound is vacuous, which is a fact that did not occur to the reviewers (including myself) who have worked on this problem for awhile. 3) The authors develop a matrix Krasulina method (original is only for learning a rank-1 subspace) and provide analysis for this method as well. They show a careful comparison with recent theoretical results and are honest about drawbacks. The weaknesses of this paper are the following, and I suggest the authors address these before presentation at NeurIPS, perhaps by moving some material or adding material to the supplement: 1) The matrix Krasulina method looks like a simple gradient method for the least squares cost. See this gradient derived in [1] given below. The authors should study the relationship of their suggested method to gradient methods more carefully. 2) Survey papers [2,3] below will be helpful for the comparison to gradient methods and for providing other references the authors may have missed. We suggest the authors do another careful lit review since this is such a well-studied area to establish connections more thoroughly. 3) The results are given as convergence in expectation, conditioned on an event Gt that will happen with high probability. The authors need to make clearer in the main text what is Gt, what random variables it depends on and what assumptions it makes. They also need to make clear what random variables the expectation is wrt to. 4) The authors should also include a comparison of the theory to the convergence results of Allen-Zhu and Li. 5) The experiments are given in iteration count, and the reviewers asked for experiments with wall clock time. Additionally, the experiments would be much stronger if other techniques are on the plots, especially VR-PCA in the synthetic data plus Oja’s in both experiments, since the paper makes extensive comparisons to these two in previous sections. [1] Balzano, Laura, Robert Nowak, and Benjamin Recht. "Online identification and tracking of subspaces from highly incomplete information." In 2010 48th Annual allerton conference on communication, control, and computing (Allerton), pp. 704-711. IEEE, 2010. [2] Comon, Pierre, and Gene H. Golub. "Tracking a few extreme singular values and vectors in signal processing." Proceedings of the IEEE 78, no. 8 (1990): 1327-1343. [3] Balzano, Laura, Yuejie Chi, and Yue M. Lu. "Streaming pca and subspace tracking: The missing data case." Proceedings of the IEEE 106, no. 8 (2018): 1293-1310.